EMBO
reports

# Rab22A recruits BLOC-1 and BLOC-2 to promote the biogenesis of recycling endosomes

Saurabh Shakya[1], Prerna Sharma[1,†], Anshul Milap Bhatt[1,†], Riddhi Atul Jani[2], Cédric Delevoye[2,3] & Subba Rao Gangi Setty[1,*] iD

## Abstract

Recycling endosomes (REs) are transient endosomal tubular intermediates of early/sorting endosomes (E/SEs) that function in cargo recycling to the cell surface and deliver the cell type-specific cargo to lysosome-related organelles such as melanosomes in melanocytes. However, the mechanism of RE biogenesis is largely unknown. In this study, by using an endosomal Rab-specific RNAi screen, we identified Rab22A as a critical player during RE biogenesis. Rab22A-knockdown results in reduced RE dynamics and concurrent cargo accumulation in the E/SEs or lysosomes. Rab22A forms a complex with BLOC-1, BLOC-2 and the kinesin-3 family motor KIF13A on endosomes. Consistently, the RE-dependent transport defects observed in Rab22A-depleted cells phenocopy those in BLOC-1-/BLOC-2-deficient cells. Further, Rab22A depletion reduced the membrane association of BLOC-1/BLOC-2. Taken together, these findings suggest that Rab22A promotes the assembly of a BLOC-1-BLOC-2-KIF13A complex on E/SEs to generate REs that maintain cellular and organelle homeostasis.

**Keywords** BLOC-1; BLOC-2; KIF13A; Rab22A; recycling endosomes
**Subject Category** Membrane & Intracellular Transport

## Introduction

The endosomal network consists of early (EE), sorting (SE), recycling (RE) and late (LE) endosomes [1]. These organelles are produced ubiquitously through the endocytic/secretory process, and they contribute to the maturation/function of lysosomes in all cells [2] and lysosome-related organelles (LROs) in specialized cells [3]. Moreover, these endosomal intermediates maintain several cellular homeostasis pathways, including those involved in secretion, signalling, nutrient/growth factor uptake, pathogen clearance, neurotransmitter storage and release [4]. The mechanism of RE biogenesis is poorly understood while the formation of EE/SE/LE has been extensively studied. Biochemical and electron microscopy (EM) studies revealed that REs originate either from E/SEs or the *trans*-Golgi network (TGN) as long tubular intermediate structures carrying cargoes or receptors towards the cell surface [1,2,5] and partly to LROs such as melanosomes in melanocytes [3]. Several Rab GTPases (Rabs), including Rab4A, 8A, 11A, 14 and 22A, have been shown to regulate the trafficking of RE containing clathrin-dependent (TfR, EGFR, ATP7A, GLUT4, etc.) and clathrin-independent (MHC1, integrin β1 receptor, β2-adrenergic receptor, etc.) endocytosed cargoes to the cell surface in non-melanocytes [6]. Moreover, Rab11A and 22A are shown to localized to these REs and maintain their structures [7,8]. However, the regulation of Rabs in the biogenesis and maintenance of REs and their targeting towards melanosomes are largely unknown.

The formation, length and stability of RE tubular structures require several cytosolic factors such as coat/adaptor proteins, kinesin motors and actin- or microtubule-remodelling proteins [9,10]. It has been shown that a multimeric protein complex, retromer in association with the WASH (Wiskott–Aldrich syndrome protein and scar homologue) complex, regulates the biogenesis of a set of REs [11]. In contrast, another set of REs is dependent on the kinesin-3 family motor KIF13A, cargo adaptor AP-1 and the endosomal eight-subunit (pallidin, muted, dysbindin, cappuccino, snapin, BLOS1, BLOS2 and BLOS3) protein complex BLOC (biogenesis of lysosome-related organelles complex)-1 in a retromer-independent fashion [8,12,13]. Interestingly, KIF13A-BLOC-1-dependent tubular REs control the trafficking of melanocyte-specific cargo like TYRP1 (tyrosinase-related protein-1) [14], ATP7A (copper transporter) [15] and OCA2 [16] to melanosomes in melanocytes and Tf/TfR (transferrin/its receptor) to the cell surface in all cell types [8]. Additionally, our earlier studies have shown that tri-subunit (HPS3, HPS5 and HPS6) protein complex BLOC-2 interacts [17] and functions downstream of BLOC-1 in these transport pathways [14,18]. Moreover, the stability of KIF13A-BLOC-1-dependent RE tubules has been shown to be dependent on the actin-cytoskeletal protein annexin A2 and ARP2/3 complex [13]. However, the membrane recruitment of BLOC-1 and BLOC-2 and their coordination with KIF13A motor in regulating the formation/function of REs remain unknown.

1 Department of Microbiology and Cell Biology, Indian Institute of Science, Bangalore, India
2 Structure and Membrane Compartments, CNRS, UMR 144, Institut Curie, PSL Research University, Paris, France
3 Cell and Tissue Imaging Facility (PICT-IBiSA), CNRS, UMR 144, Institut Curie, PSL Research University, Paris, France
*Corresponding author. Tel: +91-80-22932297/+91-80-23602301; Fax: +91-80-23602697; E-mail: subba@iisc.ac.in
†These authors contributed equally to this work

To connect the individual functions of KIF13A, BLOC-1 and BLOC-2 in regulating RE length/stability (dynamics) or biogenesis, we hypothesize a role of Rabs in coordinating these molecules on the endosomal membranes. Our small RNAi screen limited to endosomal Rabs identified Rab22A as a potential regulator of RE biogenesis. Quantitative fluorescence microscopy, live cell imaging, EM and biochemical approaches demonstrated that Rab22A localized to REs and its depletion dramatically reduced their length and number that lead to a defect in cargo delivery to the cell surface in non-melanocytic (HeLa) cells and to the melanosomes in melanocytes, concurrently causes hypopigmentation. Moreover, we found that Rab22A regulates the recruitment and stabilization of BLOC-1 and BLOC-2 on the E/SE membranes and biochemically forms a complex by associating with KIF13A motor. This Rab22A-BLOC-1-BLOC-2-KIF13A complex might maintain biogenesis of REs and loss of any of these components result in decreased RE dynamics and increased E/SE size. Overall, our study demonstrated that Rab22A acts as a key organizer in controlling the formation of REs that regulates cargo delivery to cell surface or LROs during their biogenesis.

## Results and Discussion

### Rab22A regulates RE dynamics and pigment synthesis in melanocytes

The mechanism of RE formation from E/SEs is poorly understood. Rab GTPases act as master regulators of vesicle biogenesis/transport [19]. We postulated that small Rab GTPases might also regulate the length and number of REs (referred to here as RE dynamics). REs are characterized by the localization of the SNARE STX13 (syntaxin 13) [20] and the kinesin motor KIF13A [8]. We visualized REs in HeLa cells by expressing KIF13A-YFP, which was localized (i) to enlarged E/SEs (appeared as clusters) near the cell periphery and (ii) as long tubular structures throughout the cytosol, resembling the REs [8] (Fig 1A, control sh). These KIF13A-puncta/tubular structures were also positive for RE proteins STX13, Rab11A, AP-1 and internalized Tf (Fig EV1A).

A small shRNA screen against nine endosomal Rabs showed defective KIF13A-YFP-positive REs in Rab5C-, 7A-, 9A-, 11A- and 22A-knockdown HeLa cells compared to control or Rab4A-, 5A-, 5B-, 14A-knockdown cells (Fig 1A). In these experiments, cells were transiently transfected with pooled gene-specific shRNAs, which reduced the respective Rab transcript levels equivalent to 50% or more (Fig EV1B). A similar depletion of Rab7A-, 11A- and 22A-significantly reduced the average number of KIF13A-YFP-positive tubules per cell (referred to here as $T_N$) compared to control cells. However, Rab4A-, 5B-, 5C- and 9A- depletion only moderately reduced the average $T_N$ (Fig 1B and Table 1). In addition, quantification of KIF13A-YFP-positive tubule length (referred to here as $T_L$, measured in μm) showed a significant loss in average $T_L$ in Rab11A- and 22A-knockdown and a moderate effect in Rab5C-, 7A- and 9A- depleted cells compared to control cells (Fig 1C and Table 1). These results suggest that Rab7A, 11A and 22A regulate RE dynamics either independently or cooperatively along the same pathway. We hypothesize that the modest RE defect observed in the knockdown cells of other Rabs is possibly due to the alteration in endosomal morphology.

In a recent study, we have shown that Rab9A functions in targeting REs towards melanosomes [21]. Thus, we evaluated the regulatory role of Rab9A along with Rab7A, 11A and 22A on RE dynamics in HeLa cells. At steady state, Rab7A, 9A, 11A and 22A localize equally to the STX13 or AP-1-positive subcellular membrane fractions [12] (Fig EV1C, data not shown for Rab9A) or to the STX13-labelled endosomes in HeLa cells (Fig EV1D and Table 2 for colocalization studies). However, the localization of Rab11A and 22A to the KIF13A-YFP-positive REs was moderately higher than that of Rab7A or 9A in HeLa cells (Figs 1D and EV1E). Similar to Rab11-positive tubules [8], Rab22A (tagged with GFP/mCherry) also appeared as long tubular structures resembling REs in live cell imaging experiments (Figs 1E and EV1F, and Movies EV1–EV3). Further, Rab22A-positive tubules colocalized with a cohort of Rab11A, but not with Rab7A/9A-positive vesicles (Figs 1E and EV1F, and Movies EV1–EV3). These results suggest that Rab22A-positive tubules are either generated from Rab11A-positive vesicles or represent a subpopulation of Rab11-positive RE tubules. Next, we examined the KIF13A-Rab interactions by pull-down assay using four different domains of His-tagged KIF13A (Fig 1F; motor: 1–359 aa, stalk-1: 360–800 aa, stalk-2: 801–1,306 aa and tail: 1,307–1,770 aa). Rab22A but not Rab7/9/11 showed an interaction with the stalk domain of KIF13A in HeLa cell lysate (Fig 1F). Interestingly, Rab11-KIF13A interaction has been reported previously and studied extensively using various approaches [8]. Rab11 interacts with KIF13A in live cells through a binding site encompassing likely the stalk and the tail domains of KIF13A [8]. Therefore, both Rab11 and Rab22A could associate with KIF13A at two distinct motifs. We thus tested whether Rab11A compensates the loss of Rab22A function in maintaining the KIF13A-positive REs. However, overexpressing Rab11A did not rescue the formation of long KIF13A-positive REs in Rab22A-inactivated cells (Fig EV1G), suggesting that Rab22A is required for a subpopulation of RE dynamics or biogenesis.

To test whether Rab22A regulates RE biogenesis, we expressed mCherry-Rab22A wild-type (Rab22A$^{WT}$), constitutively active (Rab22A$^{Q64L}$) and dominant negative (Rab22A$^{S19N}$) mutants along with KIF13A-YFP in HeLa cells. Both, Rab22A$^{WT}$ and Rab22A$^{Q64L}$, appeared as tubular and punctate structures that were colocalized with KIF13A-YFP (Fig 2A and Table 2 for colocalization studies). Further, the corrected total cell fluorescence (CTCF) of Rab22A$^{Q64L}$ was moderately higher than that of Rab22A$^{WT}$ (1.13-fold), suggesting an increased membrane association of Rab22A$^{Q64L}$ compared to Rab22A$^{WT}$ (Fig EV1H). Correspondingly, the CTCF of KIF13A was also enhanced (1.57-fold) in Rab22A$^{Q64L}$ compared to Rab22A$^{WT}$ expressing HeLa cells (Fig EV1H). These results indicate that Rab22A$^{Q64L}$ possibly stabilizes KIF13A association with the endosomal membranes. Consistently, mCherry-Rab22A$^{S19N}$ localized to the cytosol and abolished only the KIF13A-positive tubular structures but not the peripheral E/SEs (see below; Fig 2A). Likewise, the average $T_N$/cell and $T_L$ of KIF13A-YFP were dramatically reduced in HeLa cells expressing Rab22A$^{S19N}$ compared to Rab22A$^{WT/Q64L}$. Interestingly, these parameters showed a modest increase in cells expressing Rab22A$^{Q64L}$ mutant compared to Rab22A$^{WT}$ that was statistically insignificant (Fig 2B and C, and Table 1). These findings suggest that Rab22A regulates the $T_N$ and $T_L$ of KIF13A-positive REs similar to the MHC-1-positive tubules shown previously [7]. Additionally, subcellular membrane fractionation showed that Rab22A was enriched in the membrane (1–1.4 M sucrose) fractions positive

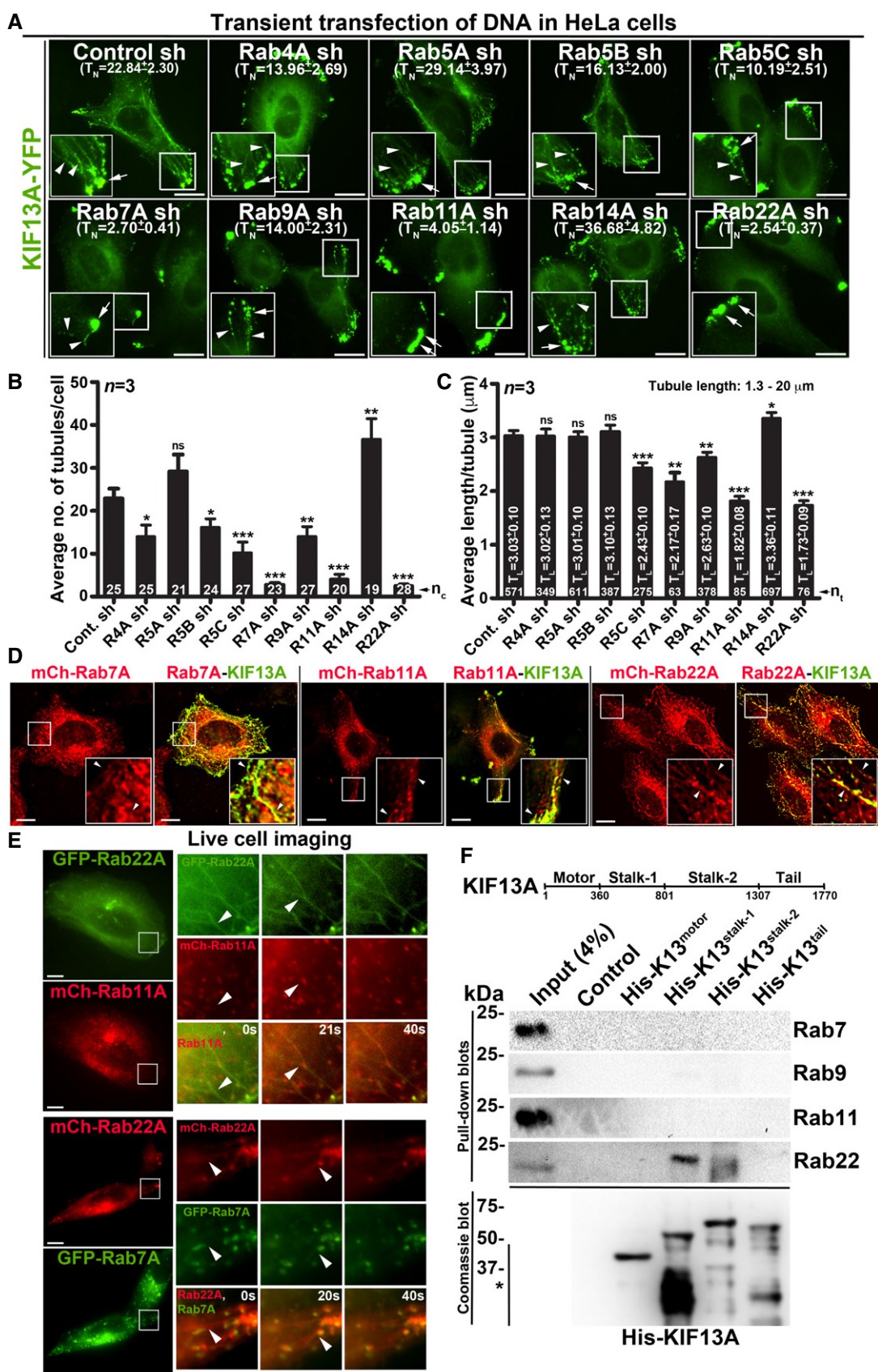

**Figure 1.**

◄ **Figure 1. Selected endosomal Rab RNAi screen identified Rab22A as a regulator of RE dynamics.**

A    IFM images of KIF13A-YFP-transfected control and Rab-knockdown HeLa cells. $T_N$: average tubule number (mean ± SEM, n = 3).

B, C  Graphs represent the measurement of KIF13A-positive $T_N$ (B) and $T_L$ (C) in HeLa cells of Fig 1A (mean ± SEM). n = 3. $n_c$: total number of cells. $n_t$: total number of tubules. *$P \leq 0.05$, **$P \leq 0.01$, ***$P \leq 0.001$ and ns = not significant (unpaired Student's t-test).

D    IFM images of KIF13A-YFP and mCherry-Rab7A/11A/22A-transfected HeLa cells.

E    Live cell imaging of GFP/mCherry-Rab22A with respect to mCherry-Rab11A or GFP-Rab7A in HeLa cells. Magnified view of insets (at 0, 20, 40 s) are shown separately.

F    Pull-down of different His-KIF13A domains using HeLa cell lysate and then probed with indicated Rab proteins. The bead-bound His-KIF13A in each pull-down was shown on the Coomassie-stained gel. *, non-specific bands. Note, part of this experiment was shown in Fig 5F.

Data information: In (A, D, E), arrowheads and arrows point to the KIF13A-/Rab22A-positive tubular REs and E/SEs, respectively. Scale bars: 10 μm.

**Table 1. Quantification of RE dynamics in HeLa cells.**

| Transfection of DNA/shRNA | Avg. no. of tubules/cell ($T_N$) | Avg. length/tubule ($T_L$ in μm) |
|---|---|---|
| Control sh | 22.84 ± 2.30 | 3.03 ± 0.10 |
| Rab4A sh | 13.96 ± 2.69 | 3.02 ± 0.13 |
| Rab5A sh | 29.14 ± 3.97 | 3.01 ± 0.10 |
| Rab5B sh | 16.13 ± 2.00 | 3.10 ± 0.13 |
| Rab5C sh | 10.19 ± 2.51 | 2.43 ± 0.10 |
| Rab7A sh | 2.70 ± 0.41 | 2.17 ± 0.17 |
| Rab9A sh | 14.00 ± 2.31 | 2.63 ± 0.10 |
| Rab11A sh | 4.05 ± 1.14 | 1.82 ± 0.08 |
| Rab14A sh | 36.68 ± 4.82 | 3.36 ± 0.11 |
| Rab22A sh | 2.54 ± 0.37 | 1.73 ± 0.09 |
| Rab22A[WT] | 24.85 ± 1.30 | 2.92 ± 0.10 |
| Rab22A[Q64L] | 26.86 ± 3.30 | 3.18 ± 0.14 |
| Rab22A[S19N] | 1.35 ± 0.25 | 1.84 ± 0.25 |
| Control sh | 24.48 ± 2.80 | 2.93 ± 0.08 |
| Rab22A sh | 3.90 ± 0.46 | 2.27 ± 0.10 |
| BLOC-1 sh | 4.39 ± 0.48 | 2.53 ± 0.09 |
| BLOC-2 sh | 3.24 ± 0.32 | 2.19 ± 0.10 |

HeLa cells were transfected with respective shRNAs or Rab22A constructs as indicated. Cells were further transfected with KIF13A-YFP, fixed with methanol and then imaged. Images were analysed for RE dynamics using Fiji Macro programme plug-in as described in Materials and Methods. Data are derived from three independent experiments (n = 3, mean ± SEM) and represented as graphs in Figs 1B and C or 2B and C or 3B and C.

**Table 2. Quantification of Pearson's coefficient values (r).**

| Cell type | Colocalization study between | r value |
|---|---|---|
| HeLa | STX13 and Rab7A | 0.75 ± 0.01 |
| | STX13 and Rab9A | 0.72 ± 0.01 |
| | STX13 and Rab11A | 0.80 ± 0.02 |
| | STX13 and Rab22A | 0.72 ± 0.03 |
| HeLa | mCherry-Rab22A[WT] and KIF13A-YFP | 0.42 ± 0.03 |
| | mCherry-Rab22A[Q64L] and KIF13A-YFP | 0.48 ± 0.05 |
| Melanocytes (melan-Ink) | mCherry-Rab22A[WT] and EEA1 | 0.30 ± 0.05 |
| | mCherry-Rab22A[WT] and TYRP1 | 0.09 ± 0.02 |
| | mCherry-Rab22A[WT] and LAMP-2 | 0.07 ± 0.01 |
| | mCherry-Rab22A[Q64L] and EEA1 | 0.31 ± 0.04 |
| | mCherry-Rab22A[Q64L] and TYRP1 | 0.16 ± 0.02 |
| | mCherry-Rab22A[Q64L] and LAMP-2 | 0.23 ± 0.03 |
| Melanocytes (melan-Ink) | Control sh: TYRP1 and EEA1 | 0.13 ± 0.01 |
| | Rab22A sh-1: TYRP1 and EEA1 | 0.40 ± 0.03 |
| | Rab22A sh-2: TYRP1 and EEA1 | 0.37 ± 0.02 |
| | Control sh: TYRP1 and LAMP-2 | 0.14 ± 0.01 |
| | Rab22A sh-1: TYRP1 and LAMP-2 | 0.45 ± 0.03 |
| | Rab22A sh-2: TYRP1 and LAMP-2 | 0.44 ± 0.02 |

The degree of colocalization between the two markers measured as Pearson's coefficient value "r" (mean ± SEM) using cellSens Dimension software (Olympus).

for the RE markers STX13 [20], KIF13A [8] and AP-1 [12] (Fig 2D). As a control, the membrane fractions were also probed for other organelle-specific makers such as GM130 (*cis*-Golgi), LIMPII (LE and TGN) and LAMP-2 (lysosome) (Fig 2D). Together, these results indicate that Rab22A localizes to REs and regulates their dynamics.

In contrast to the HeLa cells, mCherry-Rab22A[WT] in wild-type mouse melanocytes (melan-Ink) localizes to STX13-positive enlarged vacuolar endosomes [20] and also to the buds (arrowheads, persisted for < 20 s in live cell imaging) emanating from these endosomes in live cell imaging experiments. However, mCherry-Rab22A[WT] does not appear as extended tubular structures as seen in HeLa cells (Fig EV2A and Movie EV4). Consistently, mCherry-Rab22A[WT] showed colocalization with both EEA1 (EE marker) and STX13 (RE marker, data not shown), but not with TYRP1 or LAMP-2 (melanosomal or lysosomal proteins) or

melanosomes imaged in bright-field microscopy (BFM) (Fig EV2B and Table 2). The expression of mCherry-Rab22A[Q64L] mutant in melanocytes results in the formation of enlarged EEA1-positive vacuolar endosomes that were colocalized with a cohort of both TYRP1 and LAMP-2 proteins (Fig EV2B and Table 2). Whereas the cytosolic mCherry-Rab22A[S19N] mutant expression in melanocytes moderately reduced the EEA1-, LAMP-2- and TYRP1-positive organelles (Fig EV2B). These results suggest that Rab22A overexpression possibly cause a defect in cargo sorting on E/SEs. Consistent with these results, both Rab22A[Q64L]- and Rab22A[S19N]- expressing compared to Rab22A[WT]- or GFP- expressing cells showed a moderate reduction in melanocyte pigmentation without affecting the cargo (TYRP1) stability (Fig EV2B for BF images, EV2C for melanin content and EV2D for immunoblotting). Altogether, these findings indicate that RE localized Rab22A regulates cargo transport to melanosomes in melanocytes similar to cell surface trafficking of Tf/MHC-1 in HeLa cells [7,22].

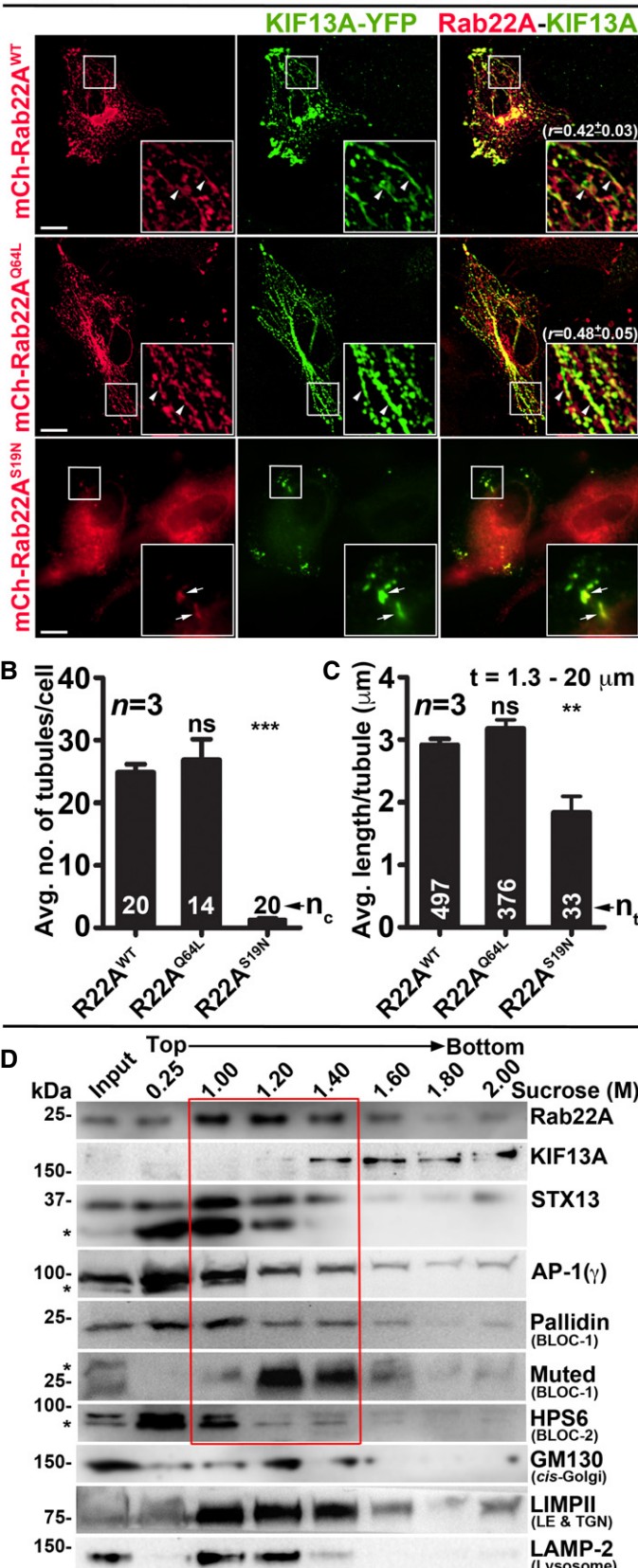

**Figure 2.**

**Figure 2. Rab22A localizes to the E/SE and REs and regulates RE dynamics.**

A IFM images of KIF13A-YFP and mCherry-Rab22A$^{WT/Q64L/S19N}$ cotransfected HeLa cells. Arrowheads and arrows point to the KIF13A-positive tubular REs and E/SEs, respectively. The colocalization coefficient ($r$, in mean $\pm$ SEM) between two markers was indicated separately. Scale bars: 10 µm.

B, C Graphs represent the measurement of KIF13A-positive $T_N$ (B) and $T_L$ (C) in HeLa cells of Fig 2A (mean $\pm$ SEM). $n = 3$. $n_c$: total number of cells. $n_t$: total number of tubules. **$P \le 0.01$, ***$P \le 0.001$ and ns = not significant (unpaired Student's $t$-test).

D Subcellular fractionation of HeLa cells to probe the localization of Rab22A (red box) with respect to other organelle-specific proteins. *, non-specific bands.

## Rab22A depletion resembles the RE defect of BLOC-1- and BLOC-2-deficient cells

The depletion of BLOC-1 or KIF13A activity in HeLa cells has been shown to cause a defect in cargo recycling and accumulation of Tf in the vacuolar endosomes [8,13]. Further, studies have shown that BLOC-2 functions downstream of BLOC-1 [14] by forming a complex [17]. However, the role of BLOC-2 in regulating RE dynamics remains unknown. Our membrane fractionation experiments showed a pool of muted/pallidin (BLOC-1 subunits) and HPS6 (BLOC-2 subunit) [14,18] co-fractionate with Rab22A (Fig 2D), indicating that BLOC-1 and BLOC-2 associate with the Rab22A-enriched membrane fractions. We reported earlier that depleting any of the BLOC-1 or BLOC-2 subunits destabilizes the integrity and function of entire BLOC complex [14,18]. Here, we knockdown muted (BLOC-1) and HPS6 (BLOC-2) subunits in HeLa cells by lentiviral transduction, confirmed their depletion by immunoblotting (see below), henceforth referred as BLOC-1$^{Mu}$ sh and BLOC-2$^{HPS6}$ sh cells. Immunofluorescence microscopy (IFM) analysis of these cells showed a dramatic reduction in KIF13A-positive RE tubules, which appeared as punctate structures similar to Rab22A-depleted cells (Fig 3A). Compared to control cells, the average $T_N$/cell and $T_L$ in both BLOC-1- and BLOC-2-deficient cells were significantly reduced, similar to that in Rab22A-knockdown cells (Fig 3B and C, and Table 1). In addition, the decreased KIF13A-RE dynamics in these cells was not due to an instability or alteration in the cytoskeletal network (microtubules were labelled anti-α-tubulin antibody and actin with phalloidin–Alexa Fluor 594; Fig EV3A). Furthermore, immunoblotting analysis revealed that Rab22A-knockdown in HeLa cells moderately reduced the stability of both BLOC-1 (muted, pallidin, dysbindin) and BLOC-2 (HPS6) subunits. In contrast, Rab22A levels in BLOC-1/BLOC-2-depleted HeLa cells were unaffected (Fig 3D). As expected, the stability of BLOC-1 subunits in BLOC-2 sh cells and BLOC-2 subunit in BLOC-1 sh cells was only modestly altered (Fig 3D). These findings suggest that membrane recruitment of Rab22A and/or BLOC-1/BLOC-2 might play a role in RE generation. Interestingly, IFM analysis showed KIF13A-positive punctate structures in Rab22A-/BLOC-1-/BLOC-2-depleted cells colocalized with AP-1 (γ subunit) [12,23], STX13 and Rab11A (Figs 3A and EV1G, data not shown for BLOC-2⁻ cells), those correspond to E/SEs. Moreover, AP-1 localization to these peripheral E/SE clusters was enhanced with a concomitant decrease in the perinuclear area in the Rab22A/BLOC-1/BLOC-2-knockdown HeLa cells compared to control cells (Fig 3A). Consistently, KIF13A, AP-1 complex, STX13 and Rab11A protein levels (Fig 3D, data not shown for Rab11) or KIF13A membrane association (Fig EV3B, data not shown for BLOC-1⁻ and BLOC-2⁻ cells) were also grossly unchanged in the knockdown cells. Together, these results indicate that KIF13A and AP-1 recruitment to the endosomal membranes is independent of Rab22A or BLOC-1/BLOC-2.

We tested whether Rab22A rescues the defective RE dynamics induced by BLOC-1- or BLOC-2 deficiency in HeLa cells. IFM experiments showed Rab22A overexpression did not generate the KIF13A-positive RE tubules; however, Rab22A did localize to the KIF13A-positive peripheral E/SEs in BLOC-1/BLOC-2-knockdown cells (Fig 3E). These results suggest that Rab22A membrane recruitment is not dependent on BLOC-1/-2, and Rab22A overexpression cannot compensate the BLOC-1/-2 deficiency in HeLa cells. Further, internalization of cargo such as Tf-Alexa Fluor 594 to the KIF13A-positive E/SEs (Fig 3A) or the fluorescein–dextran to the lysosomes (Fig EV3C, data not shown for BLOC-2⁻ cells) was unchanged in Rab22A-/BLOC-1-/BLOC-2-knockdown cells, suggesting that the endocytosis rates are not altered in these cells similar to our previous studies [14,18]. In line, the cell surface expression of LAMP-1 and M6PR (mannose 6-phosphate receptor) was also not altered in these knockdown HeLa cells (Fig EV3D, data not shown for BLOC-2⁻ cells). Quantitative IFM studies showed that Tf recycling (measured using Tf-Alexa Fluor 594) was significantly reduced in Rab22A-/BLOC-1-/BLOC-2-knockdown compared to control HeLa cells (Fig EV3E and F). The decreased Tf recycling in Rab22A-knockdown cells is likely due to the slightly reduced TfR stability compared to control cells (Fig EV3G). In contrast, the reduced Tf recycling in BLOC-1- or BLOC-2-deficient HeLa cells may not be due to TfR instability, since the protein levels were significantly elevated in these cells compared to control or Rab22A-knockdown cells (Fig EV3G). Moreover, the increased TfR levels in BLOC-1-/BLOC-2-deficient HeLa cells were similar to their respective knockout melanocytes [14,18]. Additionally, the enhanced accumulation of Tf-Alexa Fluor 594 in Rab22A-/BLOC-1-/BLOC-2-inactivated cells was not due to the decreased lysosome number or function (Fig EV3C, data not shown for BLOC-2⁻ cells). Taken together, these findings implicate that Rab22A functions in the BLOC-1-BLOC-2 pathway to regulate RE biogenesis and cargo recycling.

## Rab22A requires BLOC-1 for trafficking cargo to the melanosomes

We evaluated the role of Rab22A in melanosome protein trafficking, biogenesis and pigmentation by depleting the gene expression in wild-type melanocytes using two different shRNAs. Retroviral-mediated knockdown of Rab22A significantly reduced the melanocyte pigmentation and melanin content when compared to control cells (Fig 4A and B for bright-field images and melanin content, respectively). The decreased pigment content in Rab22A-knockdown melanocytes was comparable to the BLOC-1-deficient melanocytes (melan-mu, deficient for muted subunit; referred here as BLOC-1⁻) [14] (Fig 4B), suggesting that Rab22A might play a similar role to that of BLOC-1 complex in melanocytes. Consistently, melanosomal resident protein TYRP1 was majorly

 

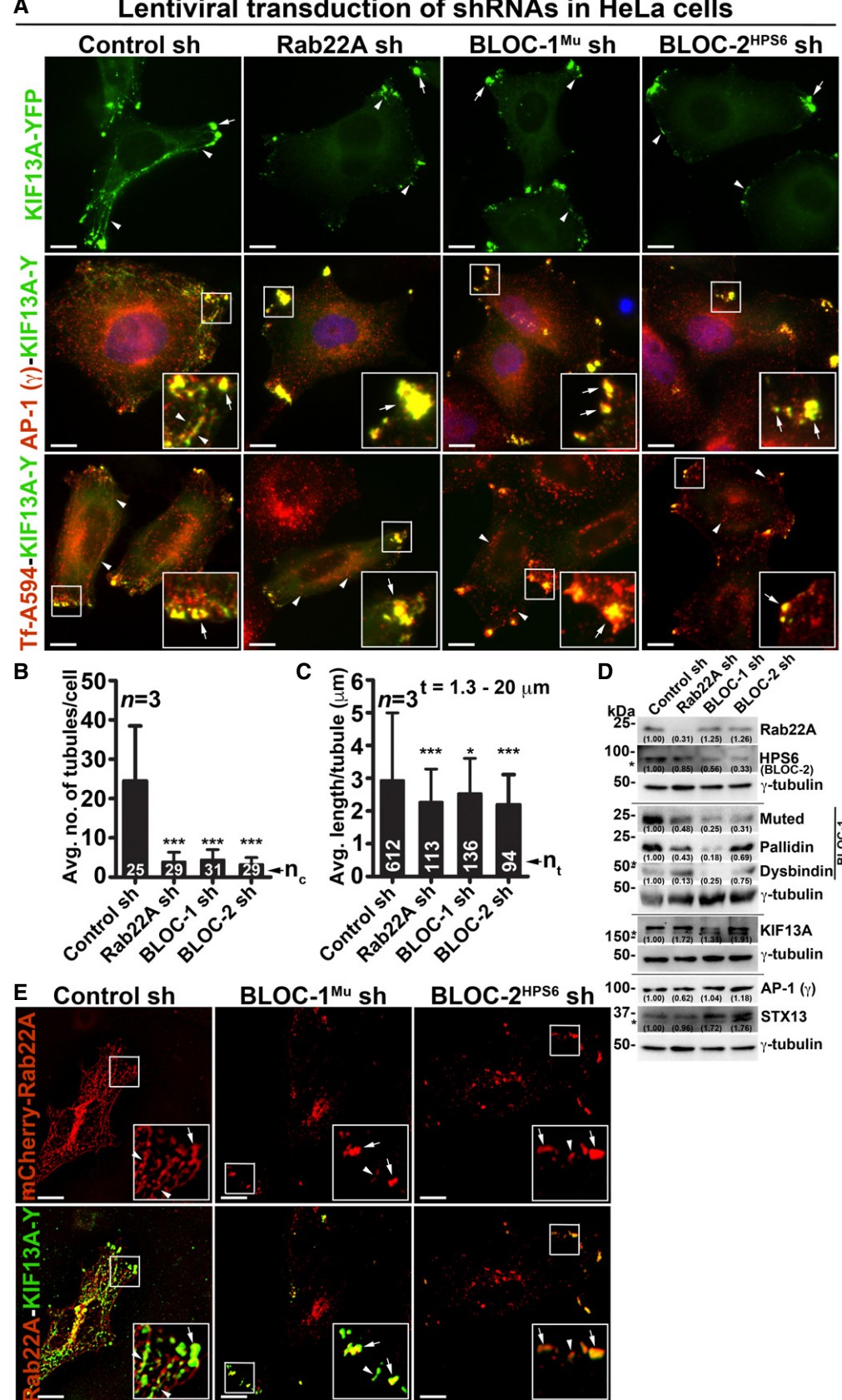

**Figure 3.**

**Figure 3.  Rab22A functions upstream in the pathway of BLOC-1 and BLOC-2 and regulates RE dynamics.**

A    IFM images of KIF13A-YFP-transfected control and Rab22A-, BLOC-1$^{Mu}$-, BLOC-2$^{HPS6}$-knockdown HeLa cells. Cells were stained for AP-1 ($\gamma$) or internalized with Tf-Alexa Fluor 594.

B, C    Graphs represent the measurement of KIF13A-positive $T_N$ (B) and $T_L$ (C) in HeLa cells of Fig 3A (mean $\pm$ SEM). $n$ = 3. $n_c$: total number of cells. $n_t$: total number of tubules. *$P \leq 0.05$ and ***$P \leq 0.001$ (unpaired Student's $t$-test).

D    Immunoblotting analysis of proteins in control and Rab22A-, BLOC-1-, BLOC-2-depleted HeLa cells. *, non-specific bands. Protein band intensities were quantified and indicated on the gels.

E    IFM images of mCherry-Rab22A and KIF13A-YFP cotransfected control and BLOC-1$^{Mu}$-, BLOC-2$^{HPS6}$-knockdown HeLa cells.

Data information: In (A, E), arrowheads and arrows point to the KIF13A-/Rab22A-positive tubular REs and E/SEs, respectively. Scale bars: 10 $\mu$m.

mislocalized to either EEA1-positive EEs or LAMP-2-positive lysosomes (for degradation) in Rab22A-depleted cells compared to control melanocytes (Fig 4A and Table 2 for colocalization studies). This was supported by live cell imaging experiments, where a cohort of TYRP1-GFP localized to the RFP-STX13-positive vacuolar endosomes in Rab22A-depleted melanocytes (Fig EV4A). As expected, TYRP1-GFP in control melanocytes localized majorly to the melanosomes while a small cohort to the RFP-STX13-positive endosomes or tubular structures (Fig EV4A) similar to other melanocytic cargoes [18,20]. Immunoblotting analyses showed that TYRP1, ATP7A and TYR (tyrosinase) protein levels were dramatically reduced, while the proteolytic processing of PMEL (full length (P1) into M$\beta$) was moderately affected in Rab22A-depleted melanocytes compared to control cells (Fig 4C). Moreover, the transcript and proteins levels of Rab22A but not Rab5 were dramatically reduced in Rab22A knockdown cells indicating that these defects are specific to Rab22A depletion in melanocytes (Figs 4C and EV4B). These findings illustrate a major defect in the cargo trafficking from E/SEs towards the melanosomes in Rab22A-inactivated cells.

The knockdown of Rab22A in MNT-1 human pigmented melanoma cells showed a noticeable loss in the number of pigmented (stage IV) melanosomes compared to control cells by EM (Fig 4D). Although the maturation of PMEL fibrils (arrowheads) was appeared as unaffected, the Rab22A-knockdown MNT-1 cells displayed stage I and stage II hybrid melanosomes corresponding to enlarged vacuolar endosomes with intraluminal PMEL fibrils (insets of Fig 4D). These findings are in line with the studies in mouse melanocytes (Fig 4A and C), suggesting that Rab22A-knockdown results in accumulation of melanocytic cargo to such hybrid organelles which then could undergo degradation. Thus, these studies indicate that Rab22A regulates the endosomal trafficking of multiple melanocytic cargoes to the melanosomes.

We examined whether Rab22A alters BLOC-1 pathway in melanocytes. Depletion of Rab22A showed slightly reduced levels of BLOC-1 (pallidin) but not of BLOC-2 (HPS6) subunits (Fig 4C) in melanocytes. In addition, overexpression of Rab22A (GFP-Rab22A$^{WT/Q64L/S19N}$) in BLOC-1$^-$ melanocytes showed no change in pigmentation or its (GFP-Rab22A$^{WT/Q64L}$) localization to STX13-positive vacuolar endosomes (Figs 4E and EV4C). Consistently, Rab22A protein levels were unchanged in the BLOC-1-deficient compared to BLOC-1-rescued melanocytes (BLOC-1$^R$) [14] while the Rab5 levels were moderately reduced in BLOC-1$^-$ melanocytes (Fig 4F). Together, these findings reaffirm that BLOC-1 is required for melanocyte pigmentation and functions downstream of Rab22A during the endosomal cargo transport to melanosomes.

## Rab22A regulates the recruitment of BLOC-1/BLOC-2 and forms endosomal complex along with KIF13A

To understand the mechanistic connection between Rab22A and BLOC-1/-2, we tested the membrane association of these molecules in Rab22A- or BLOC-1/-2-depleted cells. Subcellular membrane fractionation studies showed that the membrane association of BLOC-1 subunits (pallidin, dysbindin and muted) was dramatically reduced in Rab22A-knockdown and moderately decreased in BLOC-2$^{HPS6}$-knockdown when compared to control HeLa cells (red box, Fig 5A). Similarly, the membrane association of BLOC-2 subunit (HPS6) was notably reduced in both Rab22A- and BLOC-1$^{Mu}$-depleted cells (red box, Fig EV4D). These studies indicate that Rab22A could be required for the stabilization of both BLOC-1 and BLOC-2 association to the endosomal membranes. Consistently, membrane-cytosol fractionation experiments showed that Rab22A depletion in HeLa cells significantly reduced (approximately 28% of pallidin) the membrane association of BLOC-1 (pallidin and dysbindin subunits)

**Figure 4.  Rab22A regulates melanocyte pigmentation, cargo transport and functions upstream of BLOC-1.**

A    BF and IFM images of control and Rab22A-depleted melanocytes. The colocalization coefficient ($r$, in mean $\pm$ SEM) between the two markers indicated separately. Nuclei are stained with Hoechst 33258.

B    Graph represents the quantified melanin content in control sh, Rab22A sh and BLOC-1$^-$ melanocytes. $n$ = 3. The fold change in melanin content (mean $\pm$ SEM) indicated separately. ***$P \leq 0.001$ (unpaired Student's $t$-test).

C    Immunoblotting analysis of proteins in control sh and Rab22A sh melanocytes.

D    EM of control and Rab22A-knockdown MNT-1 melanocytes. Arrowheads indicate PMEL fibrils. Scale bars: 1 $\mu$m. I, II, III, IV: stages of melanosomes. M: mitochondria. Magnified view of insets is shown separately, and they emphasize the stage I melanosomes in both conditions. Note: the presence of hybrid stage I/II melanosomes in Rab22A-knockdown cells. Insets (b–d) and (g, h) are from different cells of respective condition. Scale bars: 200 nm.

E    BF and IFM images of GFP-Rab22A-transfected BLOC-1$^{-(Mu)}$ melanocytes.

F    Immunoblotting analysis of proteins in BLOC-1$^-$ and BLOC-1$^R$ melanocytes.

Data information: In (A, E), arrows indicate the melanocyte pigmentation and arrowheads point to the localization of TYRP1/GFP-Rab22A. Scale bars: 10 $\mu$m. In (C, F), protein band intensities were quantified and indicated on the gels. *, non-specific bands.

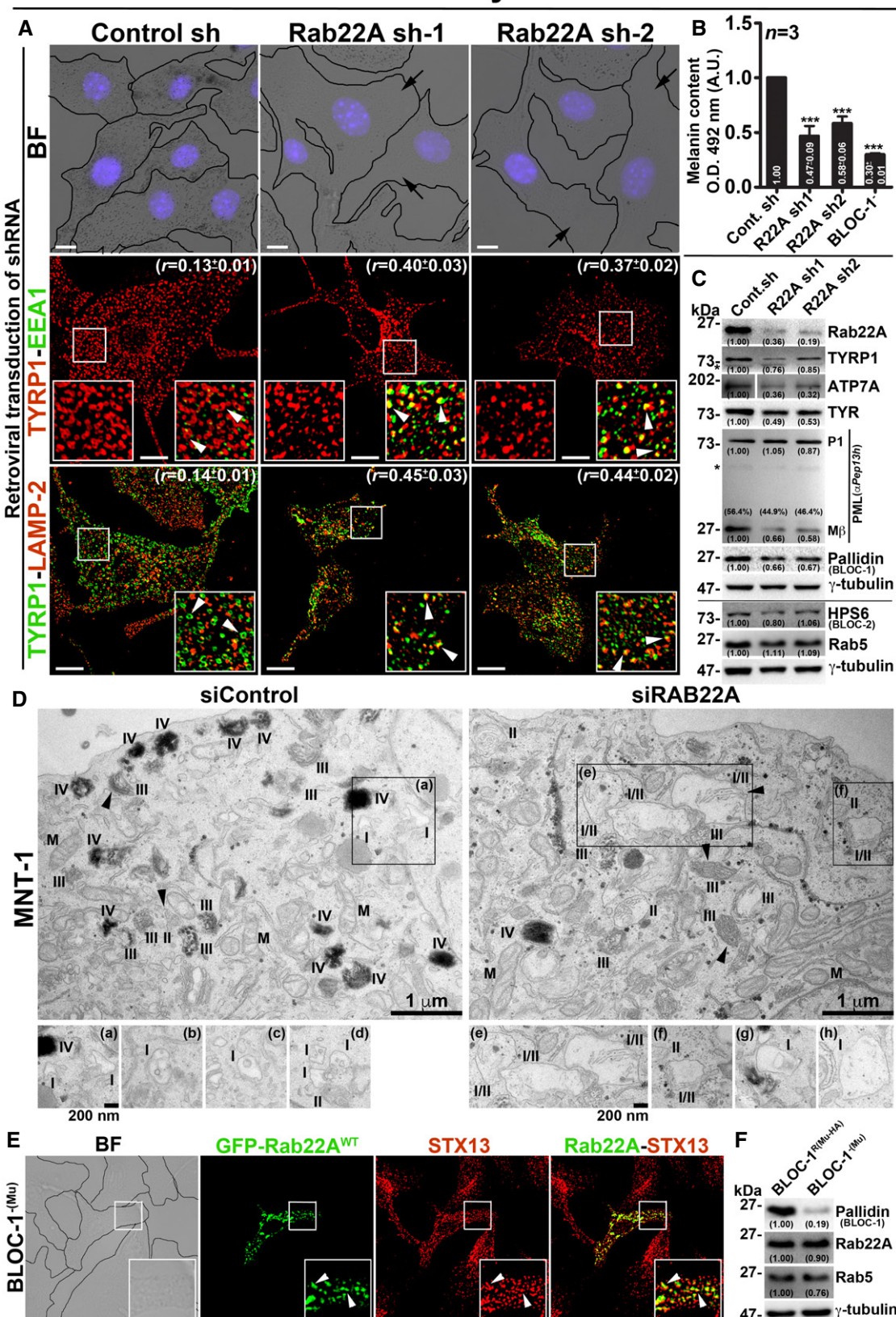

**Figure 4.**

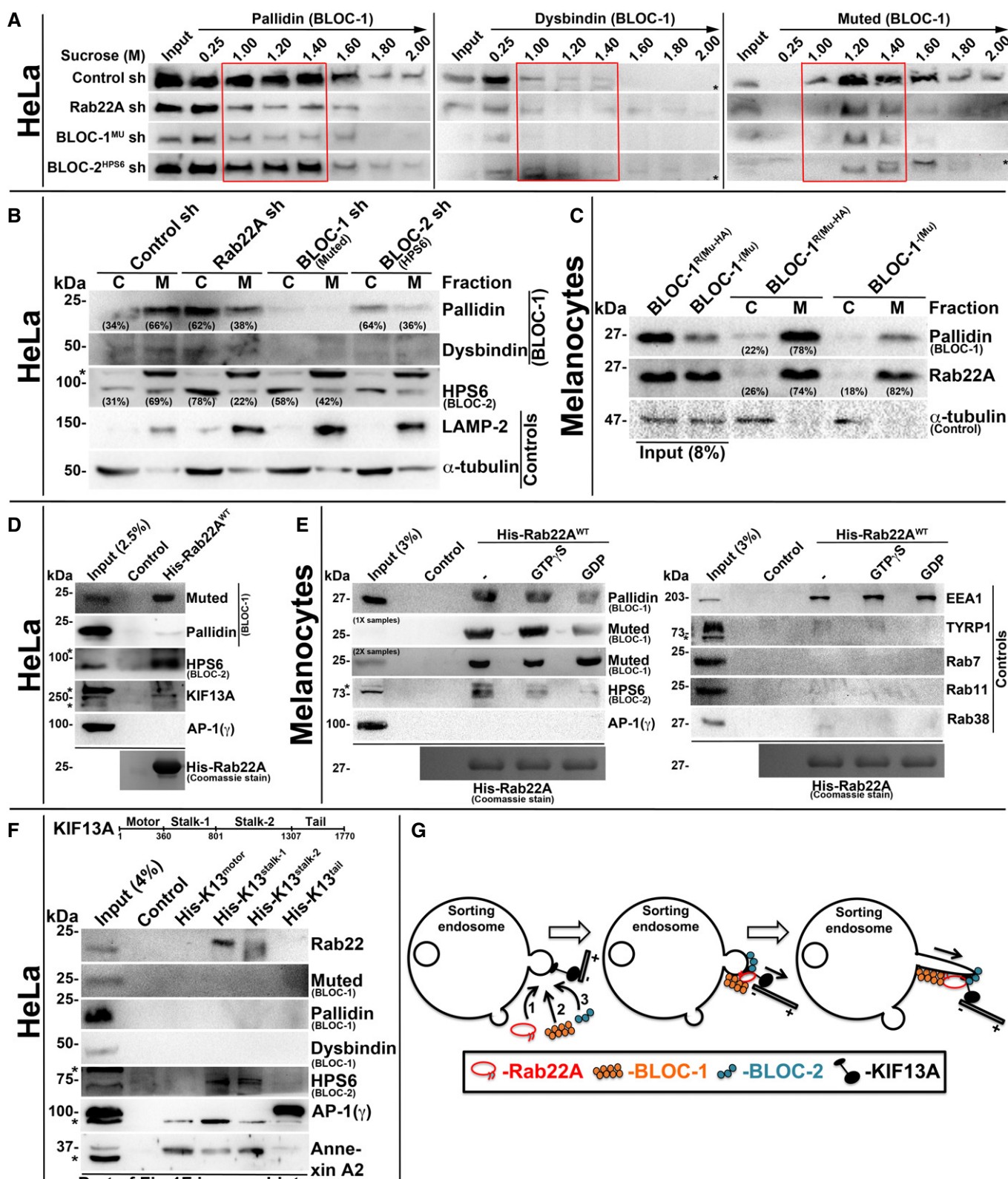

Figure 5.

(Figs 5B and EV4E). Further, Rab22A-knockdown caused a dramatic loss (approximately 47%) in BLOC-2 (HPS6 subunit) membrane association (Figs 5B and EV4E). As observed in our previous study

[17], BLOC-1 (muted subunit) depletion modestly reduced (approximately 27%) the membrane association of BLOC-2 (Figs 5B and EV4E). As expected, BLOC-2 depletion (HPS6 subunit) significantly

◄

**Figure 5.  Rab22A regulates membrane association of BLOC-1 and BLOC-2, and forms a complex with BLOC-1-BLOC-2-KIF13A.**

A    Subcellular membrane fractionation of control and Rab22A-, BLOC-1-, BLOC-2-knockdown HeLa cells and probed the fractions for pallidin, dysbindin and muted. Red coloured box emphasizes BLOC-1 membrane association in the respective cell types.

B, C   Membrane-cytosol fractionation of homogenates from HeLa cells (B) or melanocytes (C) as indicated. Protein band intensities were quantified and indicated the percentage membrane association on the gels.

D, E   Pull-down of His-Rab22A$^{WT}$ using HeLa (D) or melanocyte (E) lysate. In (E), the beads were preloaded with GTPγS or GDP.

F    Pull-down of different His-KIF13A domains using HeLa cell lysate.

G    Model illustrating the recruitment (left) and association (middle) of Rab22A, BLOC-1 and BLOC-2 in a sequential manner onto the membrane buds of E/SEs followed by interaction with KIF13A motor. Rab22A-BLOC-1-BLOC-2 complex possibly extends the membrane buds into RE tubules with KIF13A motor (right) movement on microtubules.

Data information: In (A–F), * indicates non-specific bands. In (D–F), the bead-bound His-Rab22A/His-KIF13A domains were shown on the Coomassie-stained gels separately or in Fig 1F.

altered the BLOC-1 membrane association (approximately 30% of pallidin subunit; Figs 5B and EV4E). However, Rab22A membrane association was unchanged in BLOC-1$^-$ compared to BLOC-1$^R$ melanocytes (Figs 5C and EV4F). Interestingly, these findings showed that BLOC-1 (pallidin subunit) membrane association in melanocytes (approximately 78%) is slightly higher than in HeLa cells (approximately 66%; Fig 5B and C), suggesting a possible enhanced function for BLOC-1 in melanocytes. Altogether, these experimental results suggest that Rab22A regulates the membrane recruitment or stabilization of both BLOC-1 and BLOC-2. Moreover, the association of BLOC-1 and BLOC-2, but not Rab22A, with endosomal membranes, is regulated by each other and possibly gets recruited independently of Rab22A (see below). Additionally, BLOC-1 association to the endosomal membranes possibly enhances the membrane recruitment of BLOC-2.

We next tested the interaction between Rab22A and BLOC-1/BLOC-2 complexes including the KIF13A motor. Pull-down experiments with His-tagged Rab22A$^{WT}$ using HeLa cell lysate showed that Rab22A strongly binds with BLOC-1 (muted and pallidin subunits), BLOC-2 (HPS6 subunit) and KIF13A but not with AP-1 (Fig 5D). Further, His-Rab22A displayed similar interactions with BLOC-1/BLOC-2 (but not with AP-1) in melanocyte lysate (Fig 5E), confirming that these interactions occur independent of the cell types. As expected, Rab22A showed strong interaction with EEA1 (a positive control) [24] but no binding with TYRP1 and Rab7/11/38 (negative controls; Fig 5E) in these pull-down experiments. Furthermore, GTPγS-loaded but not GDP-loaded His-Rab22A showed a strong interaction with the BLOC-1 and BLOC-2 (Fig 5E). Consistently, constitutively active His-Rab22A$^{Q64L}$ mutant reproduced this interaction in melanocyte lysates (Fig EV4G). Thus, these results revealed that Rab22A forms a complex with BLOC-1, BLOC-2 and KIF13A, but not with AP-1. However, the membrane recruitment of Rab22A and KIF13A is possibly regulated independently.

To identify the molecular mechanism between Rab22A, BLOC-1/BLOC-2, KIF13A and AP-1, we performed the pull-down experiment with the four different domains of His-tagged KIF13A (Fig 1F) using HeLa lysate. Interestingly, stalk domain of KIF13A showed an interaction with Rab22A and BLOC-2 (HPS6 subunit), but not with BLOC-1 (muted, pallidin and dysbindin) complex (Fig 5F). As previously reported, AP-1 [12] with tail domain and annexin A2 [13] with stalk and motor domains of KIF13A, respectively, showed interactions (Fig 5F). Together, these findings suggest that Rab22A bridges the BLOC-1 and BLOC-2-KIF13A in assembling the complex. Alternatively, BLOC-1 might interact with the bridging stalk domains of KIF13A protein used in these experiments and then form

complex with Rab22A-BLOC-2-KIF13A. Nevertheless, AP-1-KIF13A interaction is independent of Rab22A possibly occurs upstream along this pathway.

Overall, our study has revealed that Rab22A-BLOC-1-BLOC-2-KIF13A complex localizes to the E/SE membranes and initiates/extends the tubular structures, which eventually undergo fission and appear as recycling tubular endosomes (Fig 5G). We predict that Rab22A acts as a master regulator in these processes and explained in the following model: (i) Rab22A localizes to the endosomal buds (generated possibly by AP-1 adaptor post cargo sorting) on E/SEs where it likely recruits and/or stabilizes the association of BLOC-1 and BLOC-2 complexes onto these membranes; and (ii) Rab22A interacts further with KIF13A motor, which extends the membrane buds into tubular endosomes (Fig 5G). Our results support this model for the following reasons: (i) Rab22A localizes to the E/SEs and tubular REs; (ii) overexpression of Rab22A$^{S19N}$ (dominant negative) mutant or endogenous Rab22A depletion affects the formation of REs, concurrently increases the number of stage I/II hybrids following a block in melanosome biogenesis; (iii) individual depletion of Rab22A, BLOC-1 or BLOC-2 causes a similar cargo recycling defect and (iv) Rab22A-knockdown destabilizes the association of BLOC-1/BLOC-2 complexes to the E/SEs. Thus, our study demonstrates that Rab22A regulates the biogenesis of a set of REs that follow the slow-cargo recycling in all cells and LRO biogenesis in specialized cells. Furthermore, these Rab22A-dependent REs are different compared to the previously reported REs such as (i) MICAL-L1-syndapin2-dependent tubular REs (TREs) [25], which originate from Arf6-positive endosomes; (ii) the endocytic recycling compartments (ERC) [26], those generate from Golgi or; (iii) retromer-SNX27-dependent tubular endosomes, very likely derive from different membrane domains of E/SEs [27,28]. However, the modulation or the dynamics of these three different tubular structures by Rab22A should be evaluated in future studies. Thus, these studies suggest the existence of complexity in RE organization and cargo sorting on/to these endosomes, which is a compound problem in endosomal dynamics. Nevertheless, we assume that Rab22A-dependent REs most likely represent a fraction of total tubular RE intermediates emerged from E/SEs.

Several studies have shown that the majority of cargo recycling to cell surface is dependent on Rab11 [29]. Consistently, Rab11-knockdown in HeLa affected the dynamics of REs similar to Rab22A depletion in the cells (Fig 1 and Table 1). In contrast, overexpression of Rab11$^{S25N}$ (dominant negative) mutant in HeLa cells was shown to increase the extended tubular network, possibly by affecting the GTPase cycle [30]. However, Rab11A depletion in melanocytes moderately reduced the pigmentation compared (data

not shown) to severe hypopigmentation observed in Rab22A-inactivated cells (Fig 4A and B). Although Rab11A has been implicated in the maturation of REs [31], we hypothesize that Rab22A possibly functions downstream of Rab11 or Rab11-tubular structures morphed into a different set of REs. In the latter case, we predict that Rab11 possibly interacts with either GRAB-Rab8A [32] or Rabin8-Rab8A [33,34] proteins that were observed during the neurite outgrowth of PC12 cells. We favour the earlier model, since the overexpression of Rab11A could not rescue the KIF13A-positive tubules either in Rab22A- or BLOC-1-knockdown cells (Fig EV1G). However, this hypothesis requires further investigation.

Intriguingly, several endosomal protein complexes have been shown to regulate the function of REs: (i) the EARP (endosome-associated recycling protein) complex (Ang2-Vps52-Vps53-syndetin)—shown to regulate the recycling of internalized Tf [35]; (ii) the TOCA-CDC42-PAR-WAVE complex—shown to regulate the transport of MIG-14/Wls and TGN38 from RME-1-positive REs to TGN in *Caenorhabditis elegans* [36]; and (iii) the retriever (heterotrimer of DSCR3-C16orf62-VPS29) complex—shown to regulate the sorting and recycling of multiple cargo receptors in association with SNX17, CCC (CCDC93, CCDC22, COMMD) and WASH complexes [37]. It is unclear from these studies how many types of REs exist and how they can be distinguished from each other. Added to this complexity, multiple cargoes follow different routes, for example, copper transporter ATP7A utilizes CCC-WASH [38] or Rab22A-AP-1 molecules [39] for recycling to the cell surface and BLOC-1 towards the melanosomes [15]. Similarly, TYRP1 trafficking to melanosomes has shown to be regulated by the retromer-VARP [40] and BLOC-1-BLOC-2 [14,18] complexes, suggesting that cargo might be shared between the different types of REs. Nevertheless, our results support that Rab22A-BLOC-1-BLOC-2-KIF13A complex controls the generation (number of tubules), stabilization (length of the tubule) and likely tethering [18] of REs, but not the cargo sorting on REs. Moreover, the data presented here demonstrate that defects in function of Rab22A/BLOC-1/BLOC-2 were not compensated by any other RE-associated complexes. This suggests that these molecules and their association are important for the formation and thus the function of REs in regulating cargo recycling and LRO biogenesis. In addition, we also observed reduced expression of Rab22A equivalent to the pallidin subunit of BLOC-1 in the hippocampus of schizophrenia patient data set GSE53987 (Fig EV4H) [41], suggesting that Rab22A possibly mediates these processes in a variety of mechanisms, and it will be interesting to connect these processes to REs in future.

# Materials and Methods

## Reagents and antibodies

All chemicals and reagents were purchased either from Sigma-Aldrich (Merck) or Thermo Fisher Scientific (Invitrogen). Puromycin from Calbiochem and Matrigel from BD Biosciences were purchased. Fluorescein-conjugated dextran (D1822, IFM), Phalloidin–Alexa Fluor 594 (A12381, IFM) and Tf-Alexa Fluor 594 (T13343, IFM) were from Invitrogen.

Commercial antibodies with their specific use (IB, immunoblotting; IFM, immunofluorescence microscopy; IP, immunoprecipitation; and

FACS, fluorescence-activated cell sorting) and the catalogue numbers were indicated. Anti-dysbindin (IB, ab118795), anti-M6PR-CI (FACS, ab2733), anti-LIMPII (IB, ab16522) and anti-STX13 (IB, ab13261) were from Abcam; anti-TYRP1 (IFM, TA99) was from American Type Culture Collection (ATCC); anti-KIF13A (IB, A301-077A) was from Bethyl Laboratories; anti-γ-adaptin (AP-1; IFM and IB, 610385) and anti-GM130 (IB, 610822) were from BD Biosciences; anti-TfR (IB, H68.4) was from BioGenex; anti-EEA1 (IFM, 3288), anti-Rab5 (IB, 3547), anti-Rab9 (IB, 5118) and anti-Rab11 (IB, 5589) were from Cell Signaling Technology; anti-human LAMP-1 (FACS, H4A3), anti-human LAMP-2 (IB, H4B4) and anti-mouse LAMP-2 (IFM, GL2A7) were from Developmental Studies Hybridoma Bank; anti-GFP (IB, A-11122) was from Invitrogen; anti-T7 tag (to probe His-tag, IB, 69522) was from Novagen; anti-Rab22A (IB, 12125-1-AP), anti-Rab38 (IB, 12234-1-AP) and anti-HPS6 (IB, 11371-1-AP) were from Proteintech; anti-Annexin II (IB, sc-9061), anti-EEA1 (IB, sc-6414) and anti-TYRP1 (IB, sc-25543) were from Santa Cruz Biotechnology; anti-muted (IB, SAB1303447), anti-Rab7 (IB, R8779), anti-α-tubulin (IFM and IB, T5168) and anti-γ-tubulin (IB, T6557) were from Sigma-Aldrich. All secondary antibodies were either from Invitrogen or Jackson ImmunoResearch. The following antibodies were obtained as gift from their respective laboratories mentioned in the parenthesis: anti-STX13 (IFM and IB) [42], anti-TYR (PEP7h, IB) [43], anti-ATP7A (IB) [15], anti-muted (IB) [14], anti-HPS6 (IB) [18], anti-pallidin (IB) (Prof. Esteban Dell'Angelica, University of California, Los Angeles, USA) and anti-Pep13 h (PMEL-C-terminus, IB) (Prof. Michael S. Marks, University of Pennsylvania, Philadelphia, USA).

## Plasmids and si/shRNAs

### Bacterial expression vectors

6xHis-Rab22A-pET28a+—Rab22A$^{WT}$ and Rab22A$^{Q64L}$ genes were obtained by digesting the mCherry-Rab22A$^{WT/Q64L}$ plasmids (see below) with BglII and SalI enzymes and then subcloned into BamHI and SalI sites of pET28a+ vector (Novagen). 6xHis-KIF13A-pET28a+—various domains of human KIF13A (motor: 1–359 aa, stalk-1: 360–800 aa, stalk-2: 801–1,306 aa and tail: 1,307–1,770 aa) were PCR amplified from hKIF13AWT-YFP vector [12] and then subcloned into BamHI and HindIII sites of pET28a+ vector, which generates His-K13$^{motor}$, His-K13$^{stalk-1}$, His-K13$^{stalk-2}$ and His-K13$^{tail}$ constructs.

### Mammalian cell expression vectors

GFP/mCherry-Rab22A constructs: GFP-Rab22A$^{WT}$—mouse Rab22A was PCR amplified from cDNA derived from wild-type melan-Ink mouse melanocytes, digested and subcloned into XhoI and HindIII sites of pEGFP-C3 vector (Clontech). GFP-Rab22A$^{Q64L}$ and GFP-Rab22A$^{S19N}$—mutagenesis of Rab22A$^{WT}$ was carried out using QuickChange Multi Site-Directed Mutagenesis Kit (Agilent Technology). mCherry-Rab22A$^{WT}$, mCherry-Rab22A$^{Q64L}$ and mCherry-Rab22A$^{S19N}$—respective Rab22A fragments from pEGFP-C3 vectors were subcloned into SalI and BamHI sites of pmCherry-C1 vector (Clontech).

### Other constructs

GFP/mCherry-Rab7A—human Rab7A was PCR amplified from the cDNA obtained from HeLa cells, digested and subcloned into BglII and SalI sites of pEGFP-C1 or pmCherry-C1 vector. mCherry-Rab9A—a BamH1- and XhoI-digested fragment of human Rab9A was obtained

from Addgene vector (12263) and subcloned into BglII and SalI sites of pmCherry-C1 vector. mCherry-Rab11A—human Rab11A was PCR amplified from the cDNA obtained from HEK293T cells, digested and subcloned into BglII and SmaI sites of pmCherry-C1 vector. Human KIF13A-YFP [12], GFP-STX13-pEGFP-C1 and RFP-STX13-pCDNA3.1+ [20] have been described previously.

### Lentiviral shRNA vectors

The pooled TRC shRNA plasmids were used in the RNAi screen against Rab GTPases and also against BLOC-1 (muted subunit) and BLOC-2 (HPS6 subunit) complexes. These shRNAs were purchased from TRC genome-wide shRNA library (Sigma-Aldrich). The target sequence of each plasmid is listed in the Appendix Table S1. The knockdown efficiency of pooled shRNAs against specific Rab protein in HeLa cells was shown in Fig EV1B. Empty pLKO.1-Puro plasmid was used as a control in all shRNA-knockdown experiments (control sh). Lentiviral packaging vectors pMD2.G (VSV-G envelope, 12259) and psPAX2 (packaging, 12260) were obtained from Addgene.

### Retroviral shRNA vectors

Oligodeoxyribonucleotide duplexes containing the target sequences were cloned into the BamHI and HindIII sites of pRS shRNA vector (OriGene Technologies). The following sequences were selected as target for Rab22A-knockdown: Rab22A sh-1, 5′-AGAACGATTTC GTGCATTGGCA-3′ and Rab22A sh-2, 5′-GTCAGAGTCGTATCAGT AAGAT-3′. Empty pRS shRNA plasmid was used as a control in all shRNA-knockdown experiments (control sh). All plasmid inserts were verified by DNA sequencing.

### siRNA sequences

The sense strand for the indicated double-stranded siRNAs was synthesized with the following sequences: siRNA control (referred to here as siControl), 5′-AATTCTCCGAACGTGTCACGT-3′ and siRNA Rab22A (referred to here as siRab22A), 5′-CAGGTTTAATT TGATGGTCTA-3′.

## Cell culture, transfection and retro-/lentiviral transduction

### Human cell lines

HeLa (ATCC), HEK293T (ATCC) and PLAT-E (Cell Biolabs) were maintained in DMEM supplemented with 10% FBS (Biowest), 1% L-glutamine and 1% penicillin–streptomycin (Pen-Strep) antibiotics. MNT-1, human pigmented melanoma cell line, was grown in DMEM supplemented with 20% FBS, 10% AIM-V medium, 1% non-essential amino acids, 1% sodium pyruvate and 1% Pen-Strep antibiotics.

### Mouse melanocyte cell lines

Wild-type melan-Ink4a (from C57BL/6J *a/a Ink4a-Arf*$^{-/-}$ mice, formerly called melan-Ink4a-1 and referred to here as WT or melan-Ink4a) [44], BLOC-1-deficient melan-mu1 (from B6/CHMU/Le *Mu*$^{mu/mu}$ mice, referred to here as BLOC-1$^{-}$ or BLOC-1$^{-(Mu)}$) and BLOC-1-rescued melan-mu1 (melan-mu:MuHA, stably rescued with Mu-HA by retroviral transduction, referred to here as BLOC-1$^{R}$ or BLOC-1$^{R(MuHA)}$) [14] were maintained as described previously [45]. DNA vectors were transfected into the cells by Lipofectamine 2000 (Invitrogen) according to the manufacturer's protocol. Retroviruses

were isolated from PLAT-E cells and transduced into melanocytes as described [20]. Lentiviruses were prepared by transfecting HEK293T with mixer of DNA vector, pMD2.G (VSV-G envelope) and psPAX2 (packaging) in 1:0.2:1 ratio followed by collecting the virus at 48 and 72 h and then transduced into HeLa cells. Cells were selected once or twice with puromycin (1 or 2 μg/ml) before used it for experiments. MNT-1 cells were transfected with respective siRNAs by seeding $10^4$ cells in a 6-well plate following first transfection with siRNA (20 pmol) on 3$^{rd}$ day and repeated the second transfection on 5$^{th}$ day using Oligofectamine transfection reagent (Invitrogen). The cells were collected and processed for electron microscopy studies on 7$^{th}$ day.

## Transcript analysis (semiquantitative PCR or schizophrenia dataset)

RNA was isolated from HeLa cells or melan-Ink melanocytes using TRIzol (Sigma-Aldrich) method. Briefly, cells were treated with TRIzol reagent and then extracted with chloroform at room temperature. The aqueous layer containing RNA was precipitated with isopropanol followed by a wash with 70% ethanol. Finally, the RNA pellet was suspended in 0.01% DEPC-treated water (Sigma-Aldrich) and then estimated the concentration using NanoDrop 2000C spectrophotometer (Thermo Fisher Scientific). Further, cDNA was prepared by using cDNA synthesis kit (Fermentas). The transcripts were amplified by Bio-Rad S1000 Thermal Cycler using gene-specific primers (listed in the Appendix Table S2), and the GAPDH was used as a loading control. Band intensities were measured, normalized with GAPDH, quantified fold change with respect to the control and then listed in the figure. Transcript levels of Rab22A and pallidin (BLOC1S6, subunit of BLOC-1) in the hippocampus of schizophrenia patients were compared with normal specimens in the publicly available data set GSE53987.

## Melanin estimation

Intracellular melanin content of melanocytes was measured using a protocol as described previously [46]. Cells were transfected with GFP-Rab22A$^{WT}$, GFP-Rab22A$^{Q64L}$ or GFP-Rab22A$^{S19N}$ separately. In other experiment, cells were transduced with virus encoding control or different Rab22A shRNAs. Cells were lysed in lysis buffer [50 mM Tris–Cl pH 7.4, 2 mM EDTA, 150 mM NaCl, 1 mM DTT and 1× protease inhibitor cocktail (Sigma-Aldrich)] and then centrifuged for 15 min at 20,000 *g*, 4°C. Supernatants were subjected to protein estimation using Bradford protein estimation kit (Bio-Rad). The melanin pellet fractions were washed with ethanol:ether (1:1 ratio) mixture followed by suspending into resuspension buffer (2 M NaOH and 20% DMSO) and then incubated at 60°C for 20 min. Optical density of melanin pigments was measured at 492 nm using Tecan multi-well plate reader (Tecan) and then normalized with respective protein concentration.

## Immunoblotting

Protein levels in the cell lysate were analysed by immunoblotting as described previously [20]. Cells were harvested, lysed in RIPA buffer (50 mM Tris–Cl pH 7.4, 1% NP-40, 0.25% sodium deoxycholate, 150 mM NaCl, 1 mM EDTA, 1× protease inhibitor cocktail) and then centrifuged at 13,500 *g* for 10 min at 4°C. Protein amounts

were estimated by using Bradford reagent (Bio-Rad), and the equal amount of cell lysates was subjected to SDS–PAGE electrophoresis. Immunoblots were developed with the Clarity Western ECL substrate (Bio-Rad) and then imaged in a Molecular Imager ChemiDoc XRS+ imaging system (Bio-Rad) using Image Lab 4.1 software. Protein band intensities on the immunoblots were measured, normalized with γ-tubulin, quantified the fold change with respect to control and then indicated in the figure. % Mβ formation was calculated from the total PMEL (sum of P1 and Mβ band densities) after γ-tubulin normalization.

### Immunofluorescence microscopy and image analysis

For steady-state localization studies, cells on coverslips were fixed with methanol (KIF13A-YFP-transfected cells) or 3% formaldehyde (in PBS) and then stained with primary antibodies followed by the respective secondary antibodies as described previously [8,14]. In some experiments, cells on coverslips were subjected to internalization of Tf-Alexa Fluor 594 or fluorescein-conjugated dextran, chased for different time points, fixed with 3% formaldehyde and then imaged. Bright-field (BF) and immunofluorescence (IF) microscopy of cells was performed on an Olympus IX81 motorized inverted fluorescence microscope equipped with a CoolSNAP HQ2 (Photometrics) CCD camera using 60× (oil) U Plan super apochromat objective. Acquired images were deconvolved and analysed using cellSens Dimension software (Olympus). The colocalization between two colours was measured by selecting the entire cell excluding the perinuclear area and then estimated the Pearson's correlation coefficient ($r$) value using cellSens Dimension software. The average $r$ value from 10 to 20 cells was calculated and then represented as mean ± SEM. Note that the maximum intensity projection of undeconvolved Z-stack images was used during the measurement of $r$ values. The analysed images were assembled using Adobe Photoshop. Corrected total cell fluorescence (CTCF) of mCherry-Rab22A and KIF13A-YFP was calculated using below formula, and the mean fluorescence intensity was measured using Image J software. CTCF (in arbitrary units, A.U.) = area of the cell (mean cell fluorescence intensity-mean background fluorescence intensity). Individual and the averaged CTCF values from 6 to 9 cells were plotted separately. Length and number of KIF13A-YFP-positive tubules were quantified (listed in Table 1) in unbiased way by using below Macro programme plugged into Fiji software (ImageJ). In this analysis, images (~ 15 or more cells/condition) were captured randomly and converted their maximum intensity projections into binary and then skeletonized (2D/3D) using Fiji. Note, the tubule length was considered as 1.3–20 μm with the assumption that the size of SEs possibly ranges in the order of ≤ 1.3 μm diameter.

Macro programme: run("8-bit"); run("Tubeness", "sigma=.1935 use"); run("8-bit"); setAutoThreshold("Default dark"); //run ("Threshold..."); //setThreshold(40, 255); setOption("BlackBackground", false); run("Convert to Mask"); and run("Skeletonize").

### Live cell imaging

Cells were plated on 35-mm glass-bottomed dishes (MatTek Corporation) and then transfected with respective constructs. Post 24 h,

cells were visualized under Olympus IX81 fluorescence microscope equipped with an environmental chamber maintained at 37°C with 5% $CO_2$ and analysed by cellSens Dimension software. Time-lapse microscopy of both GFP and RFP/mCherry was performed by capturing image streams over 3–5 min using a CoolSNAP HQ2 (Photometrics) CCD camera. Images were analysed and converted into avi format for visualization.

### Conventional electron microscopy

MNT-1 cells grown on coverslips were transfected with specific siRNAs and then fixed with 2.5% glutaraldehyde in 0.1 M cacodylate buffer for 90 min on ice followed by treating the cells with 1% $OsO_4$ and 1.5% potassium ferricyanide on ice for 45 min. Cells were then subjected to ethanol dehydration, embedded in epon resin and hardened for 48 h at 60°C. The ultrathin sections were obtained by using Reichert UltracutS ultramicrotome and were stained by using uranyl acetate and lead citrate. The images were obtained by using transmission electron microscope (TEM; Tecnai Spirit G2; FEI, Eindhoven, The Netherlands) equipped with a 4k CCD camera (Quemesa, Olympus, Muenster, Germany).

### Internalization of transferrin (Tf)–Alexa Fluor 594 and fluorescein-conjugated dextran

Internalization of Tf-Alexa Fluor 594 in HeLa cells was performed as described previously [13]. Briefly, cells were starved for 30 min in serum-free medium, supplemented with 25 mM HEPES at 37°C and then incubated with complete medium with 25 mM HEPES containing Tf-Alexa Fluor 594 (30 μg/ml) for 20 min at 37°C. For studying the recycling of Tf-Alexa Fluor 594, cells were washed once with ice-cold PBS, incubated in complete medium for 0 and 40 min chase points at 37°C. Finally, coverslips were fixed with 3% formaldehyde and imaged under identical camera settings in a fluorescence microscope. Fluorescence intensity of Tf-Alexa Fluor 594 per cell was calculated using Fiji (ImageJ), normalized with 0 min time point and then plotted. Similarly, cells were incubated with 0.5 mg/ml of fluorescein-conjugated dextran (70,000 MW) beads on ice for 30 min and then chased for 2 h at 37°C. Finally, cells were washed with PBS, fixed with 3% formaldehyde and then imaged.

### Cell surface expression using flow cytometry

Cell surface expression of LAMP-1 and mannose-6-phosphate receptor (M6PR) was measured as described previously [14]. Briefly, cells were harvested, washed, suspended in ice-cold complete growth medium (supplemented 25 mM HEPES pH 7.4) containing primary antibodies: anti-LAMP1, anti-M6PR or anti-Tac (7G7.B6, ATCC, as negative control) and incubated on ice for 30–45 min. Cells were washed twice with growth medium, suspended again in growth medium containing Alexa Fluor 488-conjugated secondary antibody on ice for 30–45 min. Finally, cells were washed twice with growth medium, resuspended in FACS buffer (3% FBS, 1 mM EDTA and 0.02% sodium azide in PBS) and then analysed on FACS Canto (BD Biosciences). Mean fluorescence intensity (MFI) was calculated for LAMP-1 and M6PR proteins using FACSDiva version 6.1.3 software, normalized with Tac and then plotted.

## Membrane fractionation

### Subcellular fractionation

HeLa or melan-Ink cells were harvested, washed with ice-cold PBS and then homogenized in lysis buffer (0.25 M sucrose, 10 mM HEPES pH 7.4, 1 mM EDTA and 1× protease inhibitor cocktail) at 4°C using Dounce homogenizer. Cell lysate was clarified by centrifugation at 700 $g$ at 4°C for 10 min. Further, homogenates were fractionated on sucrose step gradient (top to bottom: 1.0, 1.2, 1.4, 1.6, 1.8 and 2.0 M sucrose buffers layered manually) by ultracentrifugation (Beckman L-80 ultracentrifuge) at 120,000 $g$ for 1 h at 4°C in SW60Ti rotor (Beckman). Fractions were collected from top to bottom and subjected to immunoblotting.

### Membrane-cytosol fractionation

This fractionation was carried out by following a slightly modified protocol from [17]. Control and knockdown HeLa cells were harvested, washed with PBS and then homogenized in buffer A (20 mM HEPES pH 7.4, 50 mM KCl, 1 mM DTT, 1 mM EGTA, 0.5 mM MgCl$_2$, 0.25 mM GTPγS and 1× protease inhibitor cocktail) at 4°C using Dounce homogenizer. Homogenates were centrifuged immediately at 15,000 $g$ for 10 min at 4°C. Supernatants (contains majorly microvesicles) were further ultracentrifuged at 120,000 $g$ for 90 min at 4°C. Finally, the supernatants were collected (cytosol fraction), and the pellets were suspended in buffer A containing 1% (w/v) Triton X-100 (membrane fraction). Similarly, BLOC-1-deficient (melan-mu) and rescued (melan-mu: MuHA) melanocytes were harvested, washed with PBS, homogenized in buffer B (10 mM HEPES pH 7.4, 250 mM sucrose, 1 mM DTT, 1 mM EGTA, 0.5 mM MgCl$_2$, 0.25 mM GTPγS, 1× protease inhibitor cocktail) and then centrifuged at 15,000 $g$ for 5 min at 4°C. Supernatants were further subjected to ultracentrifugation at 400,000 $g$ for 15 min at 4°C. Finally, supernatant (cytosol) fractions were collected, and the pellet (membrane) fractions were suspended in buffer B containing 1% (w/v) Triton X-100. For comparison, 1% Triton X-100 was added to all cytosolic fractions and then immunoblotted.

## Ni-NTA Pull-down of 6xHis-Rab22A

Different constructs of Rab22A (His-Rab22A$^{WT}$ and His-Rab22A$^{Q64L}$) and KIF13A (His-KIF13$^{motor}$, His-KIF13$^{stalk-1}$, His-KIF13$^{stalk-2}$ and His-KIF13$^{tail}$) were transformed into *Escherichia coli* BL21 (DE3) strain. As a negative control, empty pET28a+ vector was used. Transformants were induced with 1 mM isopropyl β-D-1-thiogalactopyranoside (IPTG) at an absorbance of 0.5–0.6 at 600 nm and then incubated for 4 h at 30°C. Cells were harvested, sonicated (with 2 s pulse on/off for 3–4 min) in PBS containing 1× protease inhibitor cocktail and then centrifuged at 13,500 $g$ for 10 min. Bacterial lysates were collected and incubated with equilibrated Ni-NTA beads (Invitrogen) for 2–3 h at 4°C with continuous swirling. The beads were washed five to six times with PBS and then incubated with HeLa or melanocyte cell lysates, prepared in RIPA buffer (50 mM Tris–Cl pH 7.4, 1% NP-40, 0.25% sodium deoxycholate, 150 mM NaCl, 1 mM EDTA, 1× protease inhibitor cocktail), for 6–8 h at 4°C. Beads were washed twice with wash buffer (RIPA buffer with 0.1% NP-40 and 0.025% sodium deoxycholate), suspended in 2× SDS–PAGE loading dye and immunoblotted. In some experiments, 6xHis-Rab22A$^{WT}$ was coupled with the nucleotide using a modified protocol described previously [47]. Ni-NTA bead-bound 6xHis-Rab22A$^{WT}$ was incubated with nucleotide exchange (NE) buffer (20 mM HEPES pH 7.5, 100 mM NaCl, 10 mM EDTA, 5 mM MgCl$_2$, 1 mM DTT) containing 1 mM GTPγS or GDP (Sigma-Aldrich) for 30 min at room temperature. Further, beads were washed once with nucleotide stabilization (NS) buffer (NE buffer lacking 10 mM EDTA) containing 20 µM of respective nucleotide. Finally, beads were incubated in NS buffer containing 1 mM nucleotide for 30 min at room temperature followed by washing twice with NS buffer and then use it for experiments.

## Statistical analysis

All statistical analyses were done using GraphPad Prism 5.02, and the significance was estimated by unpaired Student's $t$-test. $*P \leq 0.05$, $**P \leq 0.01$, $***P \leq 0.001$ and ns = not significant.

**Expanded View** for this article is available online.

## Acknowledgements

We thank G Raposo, G van Niel, MS Marks, E Dell'Angelica, A Peden and P Cresswell for generous gifts of reagents; EV Sviderskaya and DC Bennett for mouse melanocytes. We also thank D Majumder, D Kumari and A Ghosh for their technical help. We thank MS Reddy, DK Saini, P Rajyaguru and S Mahanty for critical reading of the manuscript. We acknowledge the PICT-IbiSA, shRNA resource center and the Bioimaging and flow cytometry facility. This work was supported by a Wellcome Trust-DBT India Alliance (500122/Z/09/Z to SRGS); CEFIPRA (4903-1 to SRGS and G Raposo); DBT-RNAi (BT/PR4982/AGR/36/718/2012 to SRGS); IISc-DBT partnership programme (to SRGS); Fondation pour la Recherche Médicale (Equipe FRM DEQ20140329491 Team label to G Raposo); Fondation ARC pour la Recherche sur le Cancer (PJA20161204965 to CD); and Institut Curie. SS was supported by IISc graduate fellowship, and RAJ was supported by CEFIPRA.

## Author contributions

SS designed and performed majority of the experiments in this study. PS and AMB carried out (IFM and pull-down) experiments required for manuscript revision. RAJ performed electron microscopy. CD provided several reagents, technical and scientific support throughout the project period. SRGS oversaw the entire project, coordinated and discussed the work with co-authors and wrote the manuscript.

## Conflict of interest

The authors declare that they have no conflict of interest.

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
