## [Review Process File · EMBO Reports]

Rab22A recruits BLOC-1 and BLOC-2 to promote the biogenesis of recycling endosomes

Saurabh Shakya¹, Prerna Sharma^{1*}, Anshul Milap Bhatt^{1*}, Riddhi Atul Jani², Cédric Delevoye^{2,3} and Subba Rao Gangi Setty^{1,†}

Review timeline:

Submission date:	15 February 2018
Editorial Decision:	26 March 2018
Additional Correspondence:	5 April 2018
Additional Correspondence:	16 April 2018
Revision received:	30 May 2018
Editorial Decision:	27 June 2018
Revision received:	2 August 2018
Editorial Decision:	15 August 2018
Revision received:	30 August 2018
Editorial Decision:	1 October 2018
Revision received:	1 October 2018
Accepted:	5 October 2018

Editor: Martina Rembold/Esther Schnapp

Transaction Report:

1st Editorial Decision

26 March 2018

Thank you for the submission of your research manuscript to EMBO reports. I apologize again for the delay in handling your manuscript but we have now received the three enclosed reports on it, which I also further discussed with the referees.

I am sorry to say, that the evaluation of your manuscript is not a positive one. As you will see, the referees raise important concerns regarding the novelty and the conclusiveness of the data. All referees point out that it was shown earlier that Rab22 functions in recycling endosome biogenesis in HeLa cells and referee 1 points out that the manuscript does not provide substantial new insight into its role in melanosome biogenesis. Moreover, both, referee 1 and 3 have a number of technical concerns and referee 1 indicated that the technical quality is low/unacceptable in the summary evaluation sheet returned with the report.

Given the nature of these concerns, the amount of work required to address them, the uncertain outcome of these experiments, and the fact that EMBO reports can only invite revision of papers that receive enthusiastic support from a majority of referees, I am sorry to say that we cannot offer to publish your manuscript.

I am sorry to disappoint you on this occasion, and hope that the referee comments will be helpful in your continued work in this area.

REFeree REPORTS

Referee #1:

In this manuscript, the authors focus on mechanisms of biogenesis of (tubular) recycling endosomes that both return internalized to cargo the plasma membrane and also, in certain cell types, direct cargo to lysosome-related organelles (such as melanosomes in melanocytes). The authors performed a limited shRNA screen for selected Rab proteins and calculated the number and length of tubular-shaped recycling endosomes as a measure of biogenesis. From this screen, the authors identified Rab22A as playing a significant role in recycling endosome biogenesis/morphology, and found that knock-down induced accumulation of cargo in endosomes or lysosomes. The authors provide evidence that Rab22A forms a complex with BLOC-1 and BLOC-2, as well as KIF13A, and that Rab22A may be required for the recruitment of the BLOC-complex proteins to the endosomal membranes.

Overall, this is an interesting manuscript that addresses important questions about the mechanism of recycling endosome biogenesis, and promotes a role for Rab22A in this process. There are, however, serious concerns at both the conceptual and technical levels. First, based on the data shown and rationale given, it is unclear why the authors focused exclusively on Rab22 (in melanosomes). Second, there is a seeming disconnect between the stated goal of the manuscript and the actual description of the study. Initially the PI maintained that recycling endosome biogenesis in melanosomes was poorly understood, but the study addresses biogenesis in HeLa cells, where considerably more is known. Indeed, a number of recent studies have addressed recycling endosome biogenesis and the various proteins and complexes involved, with emphasis on the tubular endosomes that resemble those observed by the authors for KIF13A, including studies by Takahashi et al., Wang et al., Bahl et al., Henmi et al., Xie et al., Compeer and Boes, Allison et al. Giridharan et al. and others. Even the role of Rab22A has been investigated in endosomes and recycling endosomes in non-melanocytes, undercutting, at least in part, the proposed rationale for this study. In addition to these conceptual issues, as noted below, there are a variety of concerns regarding the data and its interpretation throughout the manuscript.

It is difficult to accept the data presented in Figure 1. Rab5c is considered by the authors to cause cells to display significantly fewer tubules, with about 20% of the transcript remaining after shRNA transfection, whereas Rab5b supposedly does not display a significantly different number of tubules, and yet the expression of transcript after shRNA is at 85%. This makes no sense. In addition, KD efficacy should be measured at the protein level, and in any case, differences between Rab5a/b/c are miniscule and Rab5 protein levels should be addressed overall (same for the other Rabs).

How does Rab9 fit into the pathway? The authors previously implicate Rab9 in this pathway, but fail to discuss its role and why it differs from Rab22A. Is the pathway controlled dually by Rab22 and Rab9?

In Fig. 1D, the authors claim that the Rab22A Q64L mutant displays significantly enhanced localization to KIF13A recycling tubules, yet the numbers show a 0.48 as opposed to 0.42 Pearson's Coefficient. By what basis is this "significant?" How do the authors know that overexpressing the Rab22A mutant doesn't alter KIF13A localization?

I have concerns about the authors' use of the term RE dynamics when they are looking at fixed cells.

Figure 4A and B is not compelling. The 8% decrease in membrane association by pallidin upon Rab22A-depletion is (Fig. 4B) not convincing, and therefore, the conclusions about membrane association in Fig. 4A are similarly problematic.

Fig. 4D is not compelling. The smeared band for Muted is not convincing. How do the authors define "strongly" when they note that Rab22A strongly binds to the BLOC-1 complex?

MINOR:

Why do the control shRNA HeLa look so different in 1A and 2A for KIF13A? Can endogenous KIF13 be studied to give more uniform results? This reviewer doesn't see any tubules in the sh control, yet the average number of tubules per cell in the graph is listed at 25.

The manuscript needs to be better edited for clarity, and has numerous sentences that are written awkwardly and are difficult to follow. One example of many: "Moreover, the pigment content of the Rab22A-depleted melanocytes was almost as equal as in..."

Referee #2:

The manuscript by Shakya et al entitled "Rab22A brings BLOC-1 and BLOC-2 to promote the biogenesis of recycling endosomes" presents evidence that Rab22A, BLOC-1 and BLOC-2 are important for the formation of tubular recycling endosomes that are used for cargo recycling and delivery of cargo to melanosomes. They begin by looking for the Rab that is important for the generation of tubular recycling endosomes marked by overexpressed KIF13A. Surprisingly a number of endosomal Rabs led to a noticeable reduction of KIF13A tubules but they focused on Rab22A since it had the strongest phenotype. They show that loss of BLOC-1 or BLOC-2 also leads to loss of tubular endosomes and then show that Rab22A binds to BLOC-1, BLOC-2 and KIF13A and that Rab22A may be responsible for the recruitment of the BLOCs to the membrane. Loss of Rab22A has a pronounced phenotype on cargo transport to melanosomes, resulting in reduced pigmentation. This is an interesting finding that places Rab22A at a junction influencing endosomal membrane recycling and also traffic to the melanosome.

There are only a few issues that should be addressed (that shouldn't involve additional experiments).

1. It is mentioned in the text, p. 3 that overexpression of Rab11A did not rescue the loss of KIF13A tubules observed in Rab22A-depleted cells. Since Rab11-depleted cells also leads to loss of tubules, did Rab22A overexpression rescue this phenotype? Also, how do the authors interpret the overall loss of KIF13A tubules in many of the Rab depletion experiments? In all cases the KIF13A appears to be trapped out in the periphery.
2. The quantification of the western blots in Fig. 3C and F might be worth a second look. There is little change reported for HPS6 in C but it looks by eye to be a bigger change; the large change reported for Rab5 expression in F seems to be minor by eye.
3. Weigert et al (2004 - ref #20) was the first to report that loss of Rab22A led to loss of recycling tubules in HeLa cells, suggesting that Rab22A promoted tubule formation from SEs. This study should be cited or acknowledged in Results/Discussion.

Referee #3:

The study by Shakya and colleagues details experiments to investigate the process of formation of recycling endosomes (REs) and reports a role for Rab22a acting, at least in part, through the BLOC-1 and BLOC-2 complexes. REs are important elements of the endocytic pathway being involved in recycling proteins to the cell surface. They also contribute to trafficking of proteins to lysosome-related organelles (LROs). There is a lot known about much of the machinery involved in endosomal protein sorting although the formation of REs remains somewhat mysterious and there is certainly a case for studying how REs are formed.

The data presented in the manuscript by Shakya and colleagues does indicate that Rab22a has an important role to play in RE formation/dynamics and this represents an advance in the understanding of REs formation. I do think however that the manuscript needs substantial revision before it is suitable for publication in EMBO Reports.

Major issues:

1. The manuscript mentions at the start that REs are functionally connected to the TGN but afterwards the role of the TGN (and associated machinery) appears to be completely overlooked. AP1 is important in RE formation/function and may do so by interacting with Rab4 and Rab5 regulating proteins - could Rab22a be regulated through the same proteins that associate with AP1?

- There does appear to be some changes in AP1 localization after loss of Rab22a expression but the contribution of AP1 to RE formation does not really get much coverage in the manuscript.
2. In the introduction the authors list some of the endosomal protein sorting machinery such as retromer and the WASH complex and make distinctions between different sets of proteins. There is however ample evidence that the sets of machinery can interact indirectly or regulate each other. For example, it has been reported that the WASH complex not only associates with retromer but also with the BLOC1 complex. Another retromer accessory protein, VARP has a role in trafficking to LROs and retromer proteins have been shown to associate with proteins involved in RE tubule formation/maintenance such as EHD1. I think some changes to the introduction would be worthwhile so that the impression that is not made that the various endosomal protein sorting machineries operate in total isolation.
 3. In Figure 1, the authors describe their tests for Rab involvement in regulating RE tubule number/length as a "screen". I think it is an overstatement to call it a screen when only 8 different rab proteins were tested. What was the rationale in choosing those eight proteins? Why wasn't Rab7a, Rab8, Rab6 and Rab10, Rab21 included in the "screen?"
 4. In the tubule counting/measuring experiments, how were the cells selected for imaging? Was the imaging done blind to avoid potential bias?
 5. In figure 1G, it is hard to conclude that the proteins being investigated cofractionate when so few fractions are analyzed and no marker proteins for the TGN or sorting endosomes or lysosomes are shown on the blot. Do the fractions correspond to the steps for the 'gradient' used?
 6. In figure 2D, the quantification of the blot looks rather odd. For example, the STX13 signal in lane 3 (BLOC1sh) is apparently twice as much as lane 1 (control) but looks very similar even when you factor in the tubulin loading control. I suspect that the rather variable grey background in the different blots may have influenced the quantification. In the blot shown in figure 1, STX13 is a doublet but only a single band in Figure 2 - why so?
 7. In figure 4D or E (or both) I would like to see a coomassie stained gel of the pulldown so that it can be judged how clean/dirty the pulldowns are.

Additional Correspondence

5 April 2018

Please find the responses to your decision:

I am sorry to say, that the evaluation of your manuscript is not a positive one. As you will see, the referees raise important concerns regarding the novelty and the conclusiveness of the data.

Response: We do not agree with the Referee-1. Since, we showed the molecular mechanism in which Rab22A in association with BLOC-1-BLOC-2-KIF13A regulates the recycling endosome (RE) biogenesis. This is the first report showing a GTPase interacting with two different cytosolic complexes BLOC-1 and BLOC-2 having known function in melanocyte pigmentation and their loss in function cause Hermansky-Pudlak syndrome (an autosomal recessive disorder).

All referees point out that it was shown earlier that Rab22 functions in recycling endosome biogenesis in HeLa cells

We do not completely agree with the referees for the following reasons:

1. Rab22A has been shown to localize to the transferrin (Tf)-positive endosomal intermediates and its knockdown results in defective endocytic recycling of multiple cargoes toward cell surface (Mesa et al., 2001; Weigert et al., 2004; Magadan et al. 2006; Holloway et al. 20013 and Cebrian et al., 2016). These studies suggest that Rab22A regulates the cargo trafficking at REs but does not indicate that Rab22A is required for RE biogenesis. We have answer this important aspect in the current study.
2. Overexpression of Rab22A wild-type, constitute active and dominant negative mutants affected the Tf-positive tubular structures and resulted in enlarged endosomal structures (Weigert et al., 2004 and Magadan et al., 2006). However, none of these studies clearly showed that Rab22A promotes the biogenesis REs. In contrast, studies showed Rab22A regulates the sorting of cargo to these compartments. Our study is the first one to show that Rab22A in a complex with BLOC-1 and -2 regulates the length and number of REs, and thus cargo recycled to the cell surface/melanosomes.

and referee 1 points out that the manuscript does not provide substantial new insight into its role in melanosome biogenesis.

We do not agree with the Reviewer -1. Our MS highlights the following new insights:

1. Rab22A never been implicated in regulating the melanocyte pigmentation. This is the first study-reporting the Rab22A's role in melanogenesis.
2. Additionally, Rab22A showed an interaction with the well-known endosomal molecular complexes, BLOC-1 and BLOC-2, which regulate the pigmentation. This is the first study reporting a GTPase regulates the function of BLOC-1 and BLOC-2.
Additionally our study highlighted the following points:
3. Rab22A and Rab11 regulate the biogenesis of set of REs in which Rab22A is critically important for the dynamics of KIF13A or STX13-positive tubular REs.
4. Defects in RE dynamics observed in Rab22A-depletion phenocopied in BLOC-1 and BLOC-2-knockdown cells suggesting that BLOC-1/-2 functions downstream of Rab22A.

Moreover, both, referee 1 and 3 have a number of technical concerns

We will be able to answer/rectify all concerns (see below) during the revision and they are minor comments.

and referee 1 indicated that the technical quality is low/unacceptable in the summary evaluation sheet returned with the report.

We do not agree with the Reviwer-1. Seems to be reviewer-1 is not aware of the complexity in working with self-assembled cytosolic complexes: BLOC-1, composed of 8 subunits and BLOC-2, encodes three subunits. Studying membrane association of these complexes biochemically is very difficult due to cytosolic nature of these subunits. Moreover, only 2 antibodies against BLOC-1 and 1 antibody against BLOC-2 are available with my colleagues in the field (no good commercial antibodies are available as of today). Overexpression of any subunit of the complex may imbalance the complex architecture or its assembly on the membranes and thus, we used to detect the endogenous levels/interactions of these subunits. We will be happy to share all the raw data to the Reviwer-1 to satisfy the technical quality of the work. I request the Reviewer-1 kindly look at our MS and our responses one more time and suggest us the experiments to improve the manuscript.

Given the nature of these concerns, the amount of work required to address them, the uncertain outcome of these experiments, and the fact that EMBO reports can only invite revision of papers that receive enthusiastic support from a majority of referees, I am sorry to say that we cannot offer to publish your manuscript.

We will be able to answer or perform additional experiments to answer all the comments raised by the reviewers (including the Reviwer-1). Please allow us to revise our MS. Moreover, we request you to send our MS to the 4th reviewer in addition to Reviewer -2 and -3 to obtain their feed back on criticisms raised by Reviewer-1.

Referee #1:

In this manuscript, the authors focus on mechanisms of biogenesis of (tubular) recycling endosomes that both return internalized to cargo the plasma membrane and also, in certain cell types, direct cargo to lysosome-related organelles (such as melanosomes in melanocytes). The authors performed a limited shRNA screen for selected Rab proteins and calculated the number and length of tubular-shaped recycling endosomes as a measure of biogenesis. From this screen, the authors identified Rab22A as playing a significant role in recycling endosome biogenesis/morphology, and found that knock-down induced accumulation of cargo in endosomes or lysosomes. The authors provide evidence that Rab22A forms a complex with BLOC-1 and BLOC-2, as well as KIF13A, and that Rab22A may be required for the recruitment of the BLOC-complex proteins to the endosomal membranes.

Overall, this is an interesting manuscript that addresses important questions about the mechanism of recycling endosome biogenesis, and promotes a role for Rab22A in this process.

We thank the reviewer to appreciate our work.

There are, however, serious concerns at both the conceptual and technical levels. First, based on the data shown and rationale given, it is unclear why the authors focused exclusively on Rab22 (in melanosomes).

The rationale as follows:

1. We have chosen majority of the Rabs that are localized to endosomal compartments in RNAi screen excluding Rab7 (localizes to LEs), Rab9 (LEs and lysosomes). Moreover,

Rab7 (studies from others, Kawakami et al., 2008 and Hida et al, 2011) and Rab9 (our group, Mahanty et al., 2016) have been shown to regulate melanocyte cargo trafficking. None of the studies shown the role of Rab22A in cargo transport to melanosomes or in pigmentation.

2. We have shown recently that BLOC-1, BLOC-2 and KIF13A independently regulate the transport of melanocytic cargo from REs to melanosome during their biogenesis (Setty et al., 2007, Dennis et al., 2015 and Delevoye et al., 2009). However, how these complexes are recruited to the membranes is unknown. We predicted that any of the endosomal Rabs possibly regulate the recruitment of these molecules and thus control RE dynamics. Small shRNA screen showed Rab11A and Rab22A potentially regulate the RE dynamics. Since, Rab11A did not rescue the Rab22-dependent REs, we predicted that Rab22A is the key to RE biogenesis. Consistently, our unpublished data (not included in the MS) clearly showed KIF13A showed interaction with Rab22A but not with Rab11. Alternatively, Rab22A showed interaction with BLOC-1/-2 but not with Rab11A.

With this idea, we proceeded further with Rab22A. Currently, we are continuing our studies in evaluating the role of Rab11 and Rab14 in RE dynamics and sharing these results beyond the scope of the current MS.

Second, there is a seeming disconnect between the stated goal of the manuscript and the actual description of the study. Initially the PI maintained that recycling endosome biogenesis in melanosomes was poorly understood, but the study addresses biogenesis in HeLa cells, where considerably more is known.

We do not agree with the reviewer. Since, REs in any cell types follow the same mechanism. However, in melanocytes RE also carry melanocyte specific cargo along with non-melanocytic cargo. REs in melanocytes deliver the cargo toward melanosomes in addition to their cargo delivery to cell surface, similar to non-melanocytes. All our previous studies emphasized the importance of REs in melanosome biogenesis and pigmentation (Dennis et al., 2015, Jani et al., 2015 and Mahanty et al., 2016). Moreover, we can score the defects in RE function/ biogenesis easily in melanocytes than HeLa cells due to presence of melanin pigments. In this study we have shown the function of BLOC-1, BLOC-2 in HeLa cells, which was poorly appreciated previously. Thus, we do not have any disconnection between our goals. But we have used both HeLa and melanocytes as a model to non-melanocytes and specialized cells respectively in this study. Our data suggest that the biogenesis mechanism of REs is similar in both the cell types.

Regarding the point on biogenesis of REs in HeLa cells: We do not agree with the reviewer. There are several types of tubular endosomes depending on the type of cargo or associated proteins localized to these tubules, discussed in the MS. However, their membrane origin can be variable. The REs described both in HeLa and melanocytes in this MS are derived from sorting endosomes (SEs), where BLOC-1, BLOC-2 and KIF13A known to function. Moreover, SE endosomes also generate another type of tubules dependent on Retromer (targets Golgi or cell surface) and their mechanism of biogenesis is very well known. Here we have described distinct recycling tubules that are originated from SEs but not from Golgi.

Indeed, a number of recent studies have addressed recycling endosome biogenesis and the various proteins and complexes involved, with emphasis on the tubular endosomes that resemble those observed by the authors for KIF13A, including studies by Takahashi et al., Wang et al., Bahl et al., Henmi et al., Xie et al., Compeer and Boes, Allison et al. Giridharan et al. and others.

All recycling tubular structures morphologically looks similar to KIF13A/STX13-positive tubules. But these tubular structures are not the same. Since, most of the studies used internalized Tf as a marker to characterize these tubules. We predict that all these tubules can be labeled with internalized Tf due to the use of saturated levels of Tf in the experiments. Thus, tubules look very similar to KIF13A-positive tubules. Moreover, certain endogenous cargoes follow different types of tubules to reach their destination and this point was emphasized in the discussion of our MS.

Takahashi et al., - Study showed Rab11 localizes to REs and interacts with exocyst complex that regulates the exocytosis of REs toward plasma membrane. This study does not describe the biogenesis of REs or their origin or the mechanism.

Wang et al., - This study showed the role of Rab22 in NGF signaling and neurite outgrowth. Here Rab22A-Rabex5 interaction regulates the sorting of cargo rather than the biogenesis of RE. This study does not pointed out REs or their importance in the trafficking.

Bahl et al., - This study emphasize the EHD3 positive TRE (tubular recycling endosomes), possibly originated from Golgi and these are different than that of KIF13A-positive tubules. We have emphasized this point in the discussion of our manuscript. Our unpublished studies showed these tubular structures do not carry melanocytic cargo.

Henmi et al., We could not find the reference on PubMed.

Xie et al., - This study described the tubular recycling structures that contain cargo sorted either by CME or CIE pathway. Moreover, this paper emphasized the cargo sorting rather than the biogenesis of REs or the nature of REs.

Compeer and Boes, - These studies described the MICAL-L1-positive tubular structures similar to the TREs studied by Steve Caplan's group. We have described about these tubules in the discussion of our MS.

Allison et al.- This study describes that ESCRT-Spastin regulates the "fission" of REs rather than the biogenesis of REs.

Giridharan et al. – Several of his papers or Caplan's group showed the importance of adaptor EHD1 and their interaction with Rab8a and MICAL-L1 in endocytic recycling. All these molecules localized to the tubular structures and control either at endocytosis or tethering. We have discussed these points in the discussion.

and others

NONE OF THESE STUDIES DESCRIBED THE MECHANISM OF BIOGENESIS OF REs or how they maintained the tubular characteristics. Our study detailed the role of BLOC-1/-2 in stabilizing the tubules with the help of Rab22A.

Even the role of Rab22A has been investigated in endosomes and recycling endosomes in non-melanocytes, undercutting, at least in part, the proposed rationale for this study.

Reviewer only looking at the known role of Rab22A rather than how this Rab controls the endosome maturation into REs. None of the studies so far reported how Rab22A regulates the cargo targeting from RE to cell surface without identifying the effector proteins. Reviewer was least considerate to know the mechanism (answered here) except the published literature on Rab22A. Reviewer did not look at our new data that multisubunit complex BLOC-1 and BLOC-2 functions in the Rab22A pathway to generate the REs. Thus, our MS did not compromise for any novelty in the work and the results are additive to strengthen the function of Rab22A.

In addition to these conceptual issues, as noted below, there are a variety of concerns regarding the data and its interpretation throughout the manuscript.

We do not agree with the reviewer. If the reviewer has concern, we will be happy to provide the original data for his/her review. Our interpretations are very much suited the results of the study. If the reviewer has specific point, we will be able to justify our interpretation.

It is difficult to accept the data presented in Figure 1. Rab5c is considered by the authors to cause cells to display significantly fewer tubules, with about 20% of the transcript remaining after shRNA transfection, whereas Rab5b supposedly does not display a significantly different number of tubules, and yet the expression of transcript after shRNA is at 85%. This makes no sense.

We do not agree with reviewer that the data in Figure 1 has "no sense". Any RNAi screen, the knockdown efficiency of the genes varied depending on the potency of target sequences present in the siRNA/shRNA used. In our screen, except Rab5A and Rab5B, we have obtained more than 50% knockdown. As the reviewer might know that small ablation (either by over expression or knockdown) of Rab function can easily be scored to the phenotype. Thus, our phenotypes much stronger as we showed in Figure 1 and we have quantified for the same. We will be happy to replace the Rab5B knockdown data by repeating the experiment with most specific shRNAs. Thus, we disagree with the reviewer that entire data has "no meaning". To our surprise, the other two reviewers has no concern on the screen, why this reviewer making a big issue with the Figure 1, which gave the negative point to the editor and caused a decline from revision.

In addition, KD efficacy should be measured at the protein level, and in any case, differences between Rab5a/b/c are miniscule and Rab5 protein levels should be addressed overall (same for the other Rabs).

We agree with the reviewer that measuring protein levels in the knockdown cells provide better view in the case of isoform such as Rab5a/b/c. As reviewer pointed that there are no specific antibodies available to the Rab5 isoforms. Thus, we have shown the data in the form of transcript level, which is most acceptable form for the validation of RNAi screen. Moreover, the reviewer is making criticism more specific on Rab5 and missed the other important Rabs such as Rab11 and 22, which showed much more dramatic effect on RE dynamics. If the reviewer feels that Rab5B and 5C are not important for the study, we will be happy to remove it from the MS during revision.

How does Rab9 fit into the pathway? The authors previously implicate Rab9 in this pathway, but fail to discuss its role and why it differs from Rab22A.

We have not considered Rab9 (reduction in the number of tubules is not significant) in the current study, even though we have shown its role in cargo transport to melanosomes. To remained reviewer that we have implicated Rab9 during the fusion of REs with melanosome rather than RE biogenesis. Reviewer missed this point from our previous studies. Moreover, Rab9 in non-melanocytes majorly function between Golgi-lysosome pathway.

Is the pathway controlled dually by Rab22 and Rab9?

We do not agree with the reviewer's hypothesis. Both have different function. Our previous studies (Mahanty et al., 2016) showed Rab9 function during the fusion of REs with melanosomes with the help of Rab38/32/VARP, which regulates the STX13 and VAMP7 fusion localized on respective membranes. Where as, Rab22A (in the current study) localizes to REs and SEs but not to the melanosomes. Our data suggests that RE biogenesis from SEs possibly regulated by Rab22 and Rab11 rather than Rab9A. If the reviewer wishes to see the data, we will be happy to do the experiment and included in the revised MS.

In Fig. 1D, the authors claim that the Rab22A Q64L mutant displays significantly enhanced localization to KIF13A recycling tubules, yet the numbers show a 0.48 as opposed to 0.42 Pearson's Coefficient. By what basis is this "significant?"

We agree with the reviewer concern on Pearson's coefficient values. We would like to point to the reviewer that 'r' values are best for endocytic vesicles rather than tubular structures. Since, the intensity of the tubules, in this case, Rab22A appeared as less intense due to mCherry-tag, where as YFP-KIF13A intensity is high due to YFP fluorescence. We request the reviewer to observe the inset carefully where the intensity of Rab22A varied through out the tubule length of KIF13A. We will be happy to remove the "r" values from the MS, if needed.

How do the authors know that overexpressing the Rab22A mutant doesn't alter KIF13A localization?

We thank the reviewer for the nice question. We have not observed any changes in the localization of YFP-KIF13A to the tubular or endosomal structures in Rab22A WT or Constitutive active mutant and it was very similar to control, where we expressed GFP in the cells (data not shown). We expected Rab22A^{S19N} or Rab22A sh would effect the localization of KIF13, which we do not observe in our experiments (see Figures 1D and 2A). These results are consistent with the endogenous membrane localization of KIF13A i.e., which was not affected even in the Rab22A knockdown cells (Figure S1B). Thus, KIF13A localization is not affected with expression of Rab22A.

I have concerns about the authors' use of the term RE dynamics when they are looking at fixed cells.

For reviewer note, we have used maximum intensity projection of the Z-stack to quantify the dynamics rather than single frames of cells. Moreover, we have observed similar kinetics in live imaging (data not shown). If needed, we will be happy to provide the live images of the same.

Figure 4A and B is not compelling. The 8% decrease in membrane association by pallidin upon Rab22A-depletion is (Fig. 4B) not convincing, and therefore, the conclusions about membrane association in Fig. 4A are similarly problematic.

Reviewer should note that BLOC-1 contains 8 subunits, however, knockdown of one subunit destabilize the entire complex (Setty et al., 2007, Dennis et al., 2015). Thus, we have used Muted subunit of BLOC-1 complex for the knockdown in HeLa cells to represent the BLOC-1-deficient cells. Similarly, melan- μ melanocytes derived from Muted mice were used. However, there are no very good antibodies against the BLOC-1 subunits. Between the available 2 antibodies [anti-Muted (available with Prof. Mickey Marks, CHOP-UPENN, Philadelphia) and anti-Pallidin (Prof. Esteban Angellica, UCLA, USA)] only anti-pallidin gives specific band compared to others. Thus, we have used anti-pallidin antibody for most of the experiments. In Figure 4A, cells were knockdown for Muted subunit and measured the membrane association (using subcellular fractionation) of Pallidin in the cells. The amount of Pallidin association to the membranes is similar in both Rab22A sh and BLOC-1 sh but dramatically reduced compared to Control sh. We request the reviewer not to compare this result with Figure 4B.

In Figure 4B membrane-cytosolic fractionation was used to measure the membrane association of Pallidin as representative subunit of BLOC-1. Reviewer should note that the reduction in membrane association of pallidin in Rab22A sh is only 8%. Moreover, similar amount was observed in BLOC-2 sh cells. We have repeated the experiment many times and the reduction was around 8% to 10% with this subunit. Due to lack of good specificity with other antibodies, we could not validate these

results. With this reason, we have repeated the similar experiment in melanocytes. These experiments are very difficult to perform. Reviewer should appreciate our effort in doing these experiments rather than saying "not convinced". If the reviewer has better experimentation, suggest us, we will be able to do the experiments.

Thus, we request the reviewer to interpret the results of Figure 4A and 4B independently and consider the similarity between the experiments than the differences.

Fig. 4D is not compelling. The smeared band for Muted is not convincing.

We agree with reviewer. We will be happy to repeat the expt, and able to replace the gel.

How do the authors define "strongly" when they note that Rab22A strongly binds to the BLOC-1 complex?

The prediction is based on Fig. 4E results, where the interaction between Rab22A and Pallidin showed stronger and followed Rab nucleotide dependency.

MINOR:

Why do the control shRNA HeLa look so different in 1A and 2A for KIF13A?

The cell showed in Fig. 2A is a dividing cell and looks slightly different than Figure 1A. We will be able to replace it during revision.

Can endogenous KIF13 be studied to give more uniform results?

Antibody does not work for immunostaining other than the blotting. Thus, we expressed the motor wherever it is required.

This reviewer doesn't see any tubules in the sh control, yet the average number of tubules per cell in the graph is listed at 25.

We do not agree with the reviewer on this point. We did not observed any difference between control cells in Figure 1A or 2A. Tubules are evident in the inset also. We will be able to replace the control sh image similar to Figure 1A. We request the reviewer to look at the arrowheads indicated in the figure.

The manuscript needs to be better edited for clarity, and has numerous sentences that are written awkwardly and are difficult to follow. One example of many: "Moreover, the pigment content of the Rab22A-depleted melanocytes was almost as equal as in..."

We will be able to correct or modify it during revision.

Referee #2:

The manuscript by Shakya et al entitled "Rab22A brings BLOC-1 and BLOC-2 to promote the biogenesis of recycling endosomes" presents evidence that Rab22A, BLOC-1 and BLOC-2 are important for the formation of tubular recycling endosomes that are used for cargo recycling and delivery of cargo to melanosomes. They begin by looking for the Rab that is important for the generation of tubular recycling endosomes marked by overexpressed KIF13A. Surprisingly a number of endosomal Rabs led to a noticeable reduction of KIF13A tubules but they focused on Rab22A since it had the strongest phenotype. They show that loss of BLOC-1 or BLOC-2 also leads to loss of tubular endosomes and then show that Rab22A binds to BLOC-1, BLOC-2 and KIF13A and that Rab22A may be responsible for the recruitment of the BLOCs to the membrane. Loss of Rab22A has a pronounced phenotype on cargo transport to melanosomes, resulting in reduced pigmentation. This is an interesting finding that places Rab22A at a junction influencing endosomal membrane recycling and also traffic to the melanosome.

We thank the reviewer for the summary of our work and his/her appreciation.

There are only a few issues that should be addressed (that shouldn't involve additional experiments).

1. It is mentioned in the text, p. 3 that overexpression of Rab11A did not rescue the loss of KIF13A tubules observed in Rab22A-depleted cells. Since Rab11-depleted cells also leads to loss of tubules, did Rab22A overexpression rescue this phenotype?

We have not performed this experiment and will be able to do it during revision. Moreover, Rab11 has a known function recycling endosomes; and thus we did not focus on this Rab during experimentation.

Also, how do the authors interpret the overall loss of KIF13A tubules in many of the Rab depletion experiments?

We observed a minor phenotype showing reduced number of KIF13A-positive tubules in only few Rabs such as Rab4A, 5C, 9A (statistically non-significant, Figure 1B). We hypothesize that this mild phenotype might be due to the altered endosomal structures/organization.

In all cases the KIF13A appears to be trapped out in the periphery.

These peripheral vesicles or clusters are sorting endosomes that are positive for internalized Tf, AP-1 (Figure 2A) and STX13 (not shown).

2. The quantification of the western blots in Fig. 3C and F might be worth a second look. There is little change reported for HPS6 in C but it looks by eye to be a bigger change; the large change reported for Rab5 expression in F seems to be minor by eye.

We thank the reviewer for noticing the difference. We will be happy to replace it during the revision.

3. Weigert et al (2004 - ref #20) was the first to report that loss of Rab22A led to loss of recycling tubules in HeLa cells, suggesting that Rab22A promoted tubule formation from SEs. This study should be cited or acknowledged in Results/Discussion.

We agree with the reviewer and will include in the text during revision.

Referee #3:

The study by Shakya and colleagues details experiments to investigate the process of formation of recycling endosomes (REs) and reports a role for Rab22a acting, at least in part, through the BLOC-1 and BLOC-2 complexes. REs are important elements of the endocytic pathway being involved in recycling proteins to the cell surface. They also contribute to trafficking of proteins to lysosome-related organelles (LROs). There is a lot known about much of the machinery involved in endosomal protein sorting although the formation of REs remains somewhat mysterious and there is certainly a case for studying how REs are formed.

The data presented in the manuscript by Shakya and colleagues does indicate that Rab22a has an important role to play in RE formation/dynamics and this represents an advance in the understanding of REs formation. I do think however that the manuscript needs substantial revision before it is suitable for publication in EMBO Reports.

We thank the reviewer for the summary. We will be happy to revise the manuscript if we are allowed to resubmit it to the journal.

Major issues:

1. The manuscript mentions at the start that REs are functionally connected to the TGN but afterwards the role of the TGN (and associated machinery) appears to be completely overlooked. AP1 is important in RE formation/function and may do so by interacting with Rab4 and Rab5 regulating proteins - could Rab22a be regulated through the same proteins that associate with AP1? AP-1 is a known adaptor for cargo sorting both at the TGN and endosomes and it does through Rab4 and Rab5. Our unpublished results showed AP-1 did not show any interaction with Rab22A. But it showed a strong interaction with tail domain of KIF13A (data not shown). These results were consistent with the previous study carried out by the co-author, suggesting that AP1-KIF13A is required for the RE generation (Delevoeye et al., 2009). Thus, our study support that Rab22A may not act during the sorting of cargo rather than extending the vesicle buds into tubular structures. We have discussed about different tubular structures including the ones derived from Golgi in discussion of the MS.

There does appear to be some changes in AP1 localization after loss of Rab22a expression but the contribution of AP1 to RE formation does not really get much coverage in the manuscript.

We predict that AP-1 functions upstream in the pathway during the RE formation. Co-author of this MS already showed AP-1-KIF13A regulates the RE formation (Delevoeye et al., 2009). Studies here are more focused on the down stream of AP-1, where AP-1 and KIF13A already localized to the membranes on which Rab22A-BLOC-1-BLOC-2 promote the extension of the tubules. We have indicated this point in the discussion and due to limitation in word count (short article), we have not elaborated the AP-1 function in RE biogenesis.

2. In the introduction the authors list some of the endosomal protein sorting machinery such as retromer and the WASH complex and make distinctions between different sets of proteins. There is however ample evidence that the sets of machinery can interact indirectly or regulate each other. For example, it has been reported that the WASH complex not only associates with retromer but also with the BLOC1 complex.

We agree with the reviewer. Co-author of this MS already showed BLOC-1- WASH complex possibly required for the tubule stabilization or fission of the RE (Delevoye 2016). We predict that these molecules function down stream of Rab22A-BLOC1-BLOC-2-KIF13A complex.

Another retromer accessory protein, VARP has a role in trafficking to LROs and retromer proteins have been shown to associate with proteins involved in RE tubule formation/maintenance such as EHD1. I think some changes to the introduction would be worthwhile so that the impression that is not made that the various endosomal protein sorting machineries operate in total isolation.

We agree with the reviewer. More details of different types of tubular structures and their machinery can be included in the text. Due to word limit (short article), we were much focused in explaining the different components of our complex Rab22A-BLOC-1-BLOC-2-KIF13A. We will be happy to include different types of tubular structures and the machineries in the discussion during revision.

3. In Figure 1, the authors describe their tests for Rab involvement in regulating RE tubule number/length as a "screen". I think it is an overstatement to call it a screen when only 8 different rab proteins were tested.

We will be happy to change it to "selected Rab shRNA screen".

What was the rationale in choosing those eight proteins?

We have chosen the Rabs that are localizes to the EE, SE and RE. We have avoided late/lysosomal or melanosome localized Rabs.

Why wasn't Rab7a, Rab8, Rab6 and Rab10, Rab21 included in the "screen?"

We will be happy to perform the knockdown experiments against these Rabs. My prediction is that the indicated Rabs may also regulate different set of RE dynamics, but it will be difficult to explain at this point of time without further experiments i.e, linking these Rabs to BLOC-1/-2 complexes. This data can be included during revision.

4. In the tubule counting/measuring experiments, how were the cells selected for imaging? Was the imaging done blind to avoid potential bias?

We appreciate the reviewer point. We have taken all the cells expressing YFP-KIF13A. However, we have quantified the tubule length/number in around 15 cells/condition (except Rab14 sh) and indicated in the graph. Thus, we have followed completely blind study during the quantification.

5. In figure 1G, it is hard to conclude that the proteins being investigated cofractionate when so few fractions are analyzed and no marker proteins for the TGN or sorting endosomes or lysosomes are shown on the blot. Do the fractions correspond to the steps for the 'gradient' used?

We will be happy to include other markers as suggested during revision. The fractions are corresponds to the sucrose gradients as indicated.

6. In figure 2D, the quantification of the blot looks rather odd. For example, the STX13 signal in lane 3 (BLOC1sh) is apparently twice as much as lane 1 (control) but looks very similar even when you factor in the tubulin loading control. I suspect that the rather variable grey background in the different blots may have influenced the quantification.

We completely agree with the reviewer. We will re-quantify the blots or replaced with new blots during revision

In the blot shown in figure 1, STX13 is a doublet but only a single band in Figure 2 - why so?

We thank the reviewer for this point. We have used anti-STX13 from Andrew Peden's lab in Figure 1, where the antibody highlights two bands. In contrast, Abcam antibody detects single band used in Figure 2.

7. In figure 4D or E (or both) I would like to see a coomassie stained gel of the pulldown so that it can be judged how clean/dirty the pulldowns are.

We will be happy to provide you the full-length gel for reviewer during revision.

Additional Correspondence

16 April 2018

Thank you for your mail asking us to reconsider our decision and invite revision of your manuscript. I have meanwhile carefully read your letter and point-by-point response and I have also re-read the original referee reports. Please note that I have already discussed the reports also with the other referees before I made a final decision on your manuscript post-review. Neither referee 2 nor 3 disagreed with the report from reviewer 1. I acknowledge that referee 2 and 3 both judged the novelty of the findings differently from referee 1. I would however also like to point out that I looked at all the technical concerns from referee 1 myself and I agreed with most of them (KD efficiency in 1A, enhanced localization of Rab22A-Q64L in Fig. 1D, moderate effect on BLOC-1

membrane association in Fig. 4D). While thus two referees were overall more positive about the study, I considered the concerns from referee 1 important and valid.

I note in your letter that you would be able to address all outstanding issues raised by the three reviewers. I also note that all three reviewers judged the findings potentially interesting and I would therefore not oppose to consider a re-submission of your paper. I must however stress that the new manuscript would be treated as a new submission and would be re-evaluated at submission stage, in particular regarding the novelty of the findings. This new manuscript should nevertheless address all the reviewer's concerns in full.

Should you decide to send us a new manuscript an editorial re-evaluation would be made at the time of submission that would take into account any novel literature on the topic and importantly would assess the changes that were made to address the reviewer's criticisms. In case the submission would be assessed positively at the editorial stage, we will contact the same referees so as to provide you with a fast review and decision, however, as you'll understand I cannot promise on this point as it depends entirely on the reviewers' willingness to do so.

I am listing below again the concerns that I feel would be essential for the re-submission to be considered positively:

1) I note again that reviewer 1 was particularly concerned about the conclusions made on the role of Rab5 based on shRNA-mediated knockdown presented in Figure 1 and Figure S1B. If only 15% of the transcript level is decreased, it is difficult to make conclusions about recycling endosome biogenesis in the opinion of this referee. I looked at the respective figure after reading the report and I still agree with this concern, in particular since it remains unknown how this moderate reduction in mRNA levels affects protein levels. I thus consider it crucial to repeat the experiment with more effective shRNAs for Rab5A/B or, alternatively, to be more careful with the conclusions made. The current data are not sufficient to discriminate if the failure of the Rab5A/B knockdown to induce an effect on RE biogenesis/dynamics is truly specific (i.e., Rab5A/CB have no role) or if the residual protein is sufficient to sustain RE biogenesis.

2) Also the role of Rab9 should be discussed in more detail and how it functionally differs from Rab22. This can either be addressed with a more extensive discussion or, preferably, with experimental data.

3) It will be important to find another method to quantify the proposed enhanced localization of Rab22A-Q64L-mCherry to recycling endosomes since the current data in Fig. 1D are not sufficient to support this conclusion. Normalized fluorescent intensity could be shown instead, for example.

4) Figure 4A versus 4B: It is certainly true that the membrane association of Pallidin/BLOC-1 is hardly affected upon knockdown of Rab22A in HeLa cells in contrast to the results shown in Figure 4A. I appreciate that these two panels show the results of different experimental approaches. I also appreciate that these experiments represent a significant amount of work, but I think that more data are required to substantiate the conclusion that Rab22A regulates the recruitment of BLOC-1 to membranes.

5) All other technical and experimental concerns of referee 2 and 3 should be addressed with the exception of reviewer 3, point 3: it will be sufficient to explain the rationale for why certain Rab proteins were included in the screen and why Rab7a, Rab8, Rab10 and Rab21 were excluded. To perform further knockdown experiments will certainly yield interesting results, but these experiments have minor priority.

I am sorry that I cannot be more positive but I feel it is important to be clear on our expectations at this stage to save you from any frustrations and/or loss of time in the end. If you decide to submit a revised manuscript, please make sure to refer to our conversation in the cover letter.

Thank you for your mail asking us to reconsider our decision and invite revision of your manuscript. I have meanwhile carefully read your letter and point-by-point response and I have also re-read the original referee reports. Please note that I have already discussed the reports also with the other referees before I made a final decision on your manuscript post-review. Neither referee 2 nor 3 disagreed with the report from reviewer 1.

Thank you very much for discussing the comments of Reviewer-1 with other reviewers. Now, we have answered all of his/her comments/concerns in the revised manuscript.

I acknowledge that referee 2 and 3 both judged the novelty of the findings differently from referee 1.

We would like to thank you for recognizing this fact and we appreciate your support.

I would however also like to point out that I looked at all the technical concerns from referee 1 myself and I agreed with most of them (KD efficiency in 1A, enhanced localization of Rab22A-Q64L in Fig. 1D, moderate effect on BLOC-1 membrane association in Fig. 4D).

We have repeated several experiments and replaced the old data with the new data: **KD efficiency in 1A**: we have repeated this experiment and we have obtained 57% knockdown with Rab5A pooled shRNA and 52% knockdown with Rab5B pooled shRNA in HeLa cells. Interestingly, the localization, tubular length and number of KIF13A tubules remain similar to that of previous experiment (see Reviewer's figure for Rab5A sh, IFM data, which is similar to Fig. 1A). We have replaced the old data with new one in Fig. S1B.

enhanced localization of Rab22A-Q64L in Fig. 1D: We thank the reviewer for his/her nice suggestion. We have quantified the corrected total cell fluorescence (CTCF) in both Rab22A^{WT} and Rab22A^{Q64L} conditions and plotted, and represented as a panel in the Figure S1G. Interestingly, we observed slightly enhanced fluorescence (1.13 folds) in cells expressing Rab22A^{Q64L} (1.56×10^8 A.U.) compared to Rab22A^{WT} (1.38×10^8 A.U.) expressing cells. Likewise, we observed modestly increased fluorescence (1.57 folds) of KIF13A in Rab22A^{Q64L} (1.10×10^8 A.U.) compared to Rab22A^{WT} (0.70×10^8 A.U.) expressing cells (Fig. S1G). In line, the increased CTCF values for Rab22A^{Q64L} and KIF13A also correlate to the moderately increased Pearson's coefficient value (r) (Fig. 1F) between these molecules compared to Rab22A^{WT} and KIF13A. Thus, overexpression of Rab22A^{Q64L} slightly enhances the localization of KIF13A compared to Rab22A^{WT}. Consistently, knockdown of Rab22A showed moderately reduced localization of KIF13A to the membranes (Fig. 2A for IFM and Fig. S2B for biochemical analysis) Overall, this data suggests that overexpression of Rab22A modestly increased the membrane localization of KIF13A. We have included this data in the revised manuscript

moderate effect on BLOC-1 membrane association in Fig. 4D: We have repeated the subcellular fractionation (Fig. 4A), membrane-cytosol association studies (Fig. 4B) and pull-down experiments (Fig. 4D) and revised the entire figure. In these experiments, we have probed 3 different subunits (Pallidin, Dysbindin and Muted) of BLOC-1 (previously we probed only one subunit, Pallidin) to study the membrane association of BLOC-1 complex. The new subcellular fractionation experiment in Fig. 4A showed critically reduced levels and membrane association of all three subunits upon Rab22A-knockdown in HeLa cells. Consistently, the new membrane-cytosol association experiment in Fig. 4B showed 28% reduction in membrane association of Pallidin upon depletion of Rab22A. As expected, the membrane association of dysbindin followed the pallidin subunit (Fig. 4B), suggesting that Rab22A regulates the membrane association of BLOC-1. Additionally, Rab22A showed an interaction with BLOC-1 (pallidin and muted) subunits, shown in Fig. 4D. To identify the mechanism, we further performed KIF13A pull-down experiments (new Fig. 4G), wherein KIF13A showed an interaction with Rab22A and BLOC-2, but not with BLOC-1, suggesting that Rab22A bridges the BLOC-1 and KIF13A independently. Moreover, BLOC-2 interacts with both Rab22A and KIF13A. Based on this data, we have revised the model presented in Fig. 4H.

While thus two referees were overall more positive about the study, I considered the concerns from referee 1 important and valid.

We have addressed all the concerns of Referee 1 in this revised manuscript. Please evaluate the revised data.

I note in your letter that you would be able to address all outstanding issues raised by the three reviewers. I also note that all three reviewers judged the findings potentially interesting and I would therefore not oppose to consider a re-submission of your paper.
Many thanks for your consideration and support.

I must however stress that the new manuscript would be treated as a new submission and would be re-evaluated at submission stage, in particular regarding the novelty of the findings. This new manuscript should nevertheless address all the reviewer's concerns in full.

We have addressed all the reviewer's comments (see below) in this revision. Please re-evaluate the manuscript for further review.

Should you decide to send us a new manuscript an editorial re-evaluation would be made at the time of submission that would take into account any novel literature on the topic and importantly would assess the changes that were made to address the reviewer's criticisms.

We have incorporated the revised experimental data in the manuscript and highlighted the respective changes in the text in blue color.

In case the submission would be assessed positively at the editorial stage, we will contact the same referees so as to provide you with a fast review and decision, however, as you'll understand I cannot promise on this point as it depends entirely on the reviewers' willingness to do so.

We are fine with your opinion and final decision. We will be happy if you can get the same referees to review our revised manuscript. We are confident that the revised manuscript will change the initial opinion of referees.

I am listing below again the concerns that I feel would be essential for the re-submission to be considered positively:

Please find my answers/responses to your points.

1) I note again that reviewer 1 was particularly concerned about the conclusions made on the role of Rab5 based on shRNA-mediated knockdown presented in Figure 1 and Figure S1B. If only 15% of the transcript level is decreased, it is difficult to make conclusions about recycling endosome biogenesis in the opinion of this referee. I looked at the respective figure after reading the report and I still agree with this concern, in particular since it remains unknown how this moderate reduction in mRNA levels affects protein levels. I thus consider it crucial to repeat the experiment with more effective shRNAs for Rab5A/B or, alternatively, to be more careful with the conclusions made.

We have repeated these experiments more carefully. Now, we have obtained 57% knockdown with Rab5A pooled shRNA and 52% knockdown with Rab5B pooled shRNA, similar to other Rabs. Further, the localization and tubule length and number of KIF13A were not affected in Rab5A or 5B depleted cells compared to control cells (see Reviewer's figure for IFM data, similar to Fig. 1A). We have revised the Fig. S1B.

The current data are not sufficient to discriminate if the failure of the Rab5A/B knockdown to induce an effect on RE biogenesis/dynamics is truly specific (i.e., Rab5A/CB have no role) or if the residual protein is sufficient to sustain RE biogenesis.

The revised data clearly indicates that knockdown of either Rab5A or 5B had no effect on RE biogenesis. In contrast, Rab5C-knockdown moderately affected the tubule number and length, but not as significant as Rab7A or 11A or 22A-depletion in HeLa cells. Note, we observed enhanced cell death with stringent selection of Rab5A/5B/5C shRNAs (possibly increase the knockdown efficiency) and the cells are not in a good/healthy condition to use it for any experiments.

2) Also the role of Rab9 should be discussed in more detail and how it functionally differs from Rab22. This can either be addressed with a more extensive discussion or, preferably, with experimental data.

We have performed several experiments to determine the role of Rab9A in RE biogenesis: we have evaluated (1) the localization of Rab9A to sorting/recycling endosomes by IFM (Fig. S1D); (2) effect of Rab9A expression on KIF13A-tubular structures by IFM (Fig. S1E); (3) localization of

Rab9A relative to Rab22A by using live cell imaging (Fig. S1F), and (4) interaction between Rab9 and KIF13A using pull-down assay (Fig. 4G). These results clearly demonstrated that Rab9A is not associated with Rab22A-BLOC-1-BLOC-2-KIF13A pathway. Consistently, Rab9A-knockdown had no effect on RE tubule length, but moderately affected the tubule number in HeLa cells. This data is further consistent with its localization to the sorting endosomes (Fig. S1D). We hypothesized that Rab9A possibly functions either during fusion of REs with the melanosomes (Mahanty et al., *Pigment. Cell Melanoma Res.*, 2016) or during endo-lysosome fusion in non-pigmented cells. Thus, Rab9A functionally different from Rab22A in regulating the RE dynamics. These points were included in the revised manuscript.

3) It will be important to find another method to quantify the proposed enhanced localization of Rab22A-Q64L-mCherry to recycling endosomes since the current data in Fig. 1D are not sufficient to support this conclusion. Normalized fluorescent intensity could be shown instead, for example.

We appreciate your wonderful suggestion. We have quantified the CTCF (corrected total cell fluorescence) in both Rab22A^{WT} and Rab22A^{Q64L} conditions and plotted, and represented as an additional panel in the Figure S1G. Interestingly, we observed slightly enhanced fluorescence (1.13 folds) in cells expressing Rab22A^{Q64L} (1.56×10^8 A.U.) compared to Rab22A^{WT} (1.38×10^8 A.U.) expressing cells. Likewise, we observed modestly increased fluorescence (1.57 folds) of KIF13A in Rab22A^{Q64L} (1.10×10^8 A.U.) compared to Rab22A^{WT} (0.70×10^8 A.U.) expressing cells (Fig. S1G). In line, the increased CTCF values for Rab22A^{Q64L} and KIF13A also correlate to the moderately increased Pearson's coefficient value (r) (Fig. 1F) between these molecules compared to Rab22A^{WT} and KIF13A. Thus, overexpression of Rab22A^{Q64L} slightly enhances the localization of KIF13A compared to Rab22A^{WT}. Consistently, knockdown of Rab22A showed moderately reduced localization of KIF13A to the membranes (Fig. 2A for IFM and Fig. S2B for biochemical analysis). Overall, this data suggests that overexpression of Rab22A modestly increased the membrane localization of KIF13A. We have included this data in the revised manuscript.

4) Figure 4A versus 4B: It is certainly true that the membrane association of Pallidin/BLOC-1 is hardly affected upon knockdown of Rab22A in HeLa cells in contrast to the results shown in Figure 4A. I appreciate that these two panels show the results of different experimental approaches. I also appreciate that these experiments represent a significant amount of work, but I think that more data are required to substantiate the conclusion that Rab22A regulates the recruitment of BLOC-1 to membranes.

Thanks for your comments on BLOC-1 association in Rab22A knockdown cells. To evaluate the role of Rab22A in regulating the BLOC-1 membrane association, we extensively searched for antibodies against different subunits of BLOC-1. Luckily, we found two new antibodies: anti-dysbindin (from Abcam, ab118795) and anti-Muted (from Sigma-Aldrich, SAB1303447) and evaluated them using respective knockdown cell lysates (data not shown) and standardized the conditions. These antibodies work at very low dilution with minimal non-specificity. We have repeated majority of the experiments and probed with these antibodies, and included the data at respective places.

Using these antibodies, we have repeated the subcellular fractionation (Fig. 4A), membrane-cytosol association studies (Fig. 4B) and pull-down experiments (Fig. 4D) and revised the entire Figure 4. In these experiments, we probed for at least two different subunits (Pallidin, Dysbindin or Muted) of BLOC-1 (previously we probed only one subunit, Pallidin) to study the membrane association of BLOC-1 complex. The new subcellular fractionation experiment in Fig. 4A showed all three subunits were critically reduced their levels and membrane association upon Rab22A-knockdown in HeLa cells. Consistently, the new membrane-cytosol association experiment in Fig. 4B showed 28% reduction in membrane association of Pallidin upon depletion of Rab22A. As expected, the membrane association of dysbindin followed the pallidin subunit (Fig. 4B), suggesting that Rab22A regulates the membrane association of BLOC-1. Additionally, Rab22A showed an interaction with BLOC-1 (pallidin and muted) subunits, shown in Fig. 4D. To identify the mechanism, we further performed KIF13A pull-down experiments (Fig. 4G), wherein KIF13A showed an interaction with Rab22A and BLOC-2, but not with BLOC-1, suggesting that Rab22A bridges the BLOC-1 and KIF13A independently. Moreover, BLOC-2 interacts with both Rab22A and KIF13A. Based on this

data, we have revised the model and present in Fig. 4H. Thus, the results of revised Fig. 4A and 4B are consistent with each other.

5) All other technical and experimental concerns of referee 2 and 3 should be addressed with the exception of reviewer 3, point 3: it will be sufficient to explain the rationale for why certain Rab proteins were included in the screen and why Rab7a, Rab8, Rab10 and Rab21 were excluded. To perform further knockdown experiments will certainly yield interesting results, but these experiments have minor priority.

We have addressed all the technical points raised by the referees 2 and 3. Few of their points were addressed as new Reviewer's figure attached at the end of this document.

Regarding the point 3 of reviewer-3: we have evaluated the role of Rab7A, Rab8, Rab10 and Rab21 in regulating the RE dynamics. We have enclosed this preliminary data in the Reviewer's figure (see Reviewer-3, Q7). Interestingly, we found Rab7A, but not Rab8, 10, 21-knockdown severely affected the KIF13A-positive tubules. Thus, Rab8, 10 and 21 (except Rab7A) may not function in regulating RE dynamics. We observed, RE tubules in Rab7A-depleted cells were similar to Rab11A sh and Rab22A sh and thus, we have added this data to Fig. 1A, 1B and 1C. Further, we have evaluated the role of Rab7A in RE dynamics by performing the following experiments: (1) the localization of Rab7A to sorting/recycling endosomes by IFM and subcellular fractionation (Fig. S1C and S1D); (2) effect of Rab7A expression on KIF13A-tubular structures by IFM (Fig. 1D); (3) localization of Rab7A relative to Rab22A by live cell imaging (Fig. 1E); and (4) interaction between Rab7 and KIF13A/Rab22A using pull-down assay (Fig. 4E and 4G). These results clearly demonstrated that Rab7A is not associated with Rab22A-BLOC-1-BLOC-2-KIF13A pathway.

I am sorry that I cannot be more positive but I feel it is important to be clear on our expectations at this stage to save you from any frustrations and/or loss of time in the end. If you decide to submit a revised manuscript, please make sure to refer to our conversation in the cover letter.

Thanks so much Dr. Martina for your support and extensive review. We are very happy that the new data strengthen our manuscript further without changing our original idea. Moreover, we have extended our study to demonstrate the mechanism of Rab22A regulation on BLOC-1, BLOC-2 and KIF13A.

Wishing you success with the work ahead, Martina - Thanks so much!

Conversation-2: Initial decision and review comments on our Manuscript EMBOR-2018-45918V1

Dear Dr. Martina,

We thank the reviewers for their critical review/suggestions. However, we do not agree with the criticisms made by the Reviewer-1 and most of his/her comments are not acceptable to us (see below). However, we have answered all the concerns raised by the 3 reviewers (see below, point-by-point responses) in this revised manuscript. Thus, I request you to review our revised manuscript at your esteemed journal.

Please find the responses to your initial decision:

Editor P1: I am sorry to say, that the evaluation of your manuscript is not a positive one. As you will see, the referees raise important concerns regarding the novelty and the conclusiveness of the data.

Response: We do not agree with the Referee-1. Since, we showed the molecular mechanism in which Rab22A (but not Rab7, Rab9A or Rab11A) in association with BLOC-1-BLOC-2-KIF13A regulates the recycling endosome (RE) biogenesis. This is the first report showing a GTPase interacting with two different cytosolic complexes BLOC-1 and BLOC-2 having known function in melanocyte pigmentation and their loss in function cause Hermansky-Pudlak syndrome (an autosomal recessive disorder).

Editor P2: All referees point out that it was shown earlier that Rab22 functions in recycling endosome biogenesis in HeLa cells

Response: We do not completely agree with the referees for the following reasons:

3. Rab22A has been shown to localize to the transferrin (Tf)-positive endosomal intermediates and its knockdown results in defective endocytic recycling of multiple cargoes toward cell surface (Mesa et al., 2001; Weigert et al., 2004; Magadan et al. 2006; Holloway et al. 20013 and Cebrian et al., 2016). These studies suggest that Rab22A regulates the cargo trafficking at REs but do not indicate that Rab22A is required for RE biogenesis. We have answered this important aspect in the current study.
4. Overexpression of Rab22A wild type, constitutive active and dominant negative mutants affected the Tf-positive tubular structures and resulted in enlarged endosomal structures (Weigert et al., 2004 and Magadan et al., 2006). However, none of these studies clearly showed that Rab22A promotes the biogenesis REs. In contrast, studies showed Rab22A regulates the sorting of cargo to these compartments. Our study is the first one to show that Rab22A in a complex with BLOC-1 and -2 regulates the length and number of REs, and thus cargo recycled to the cell surface/melanosomes.

Editor P3: and referee 1 points out that the manuscript does not provide substantial new insight into its role in melanosome biogenesis.

Response: We do not agree with the Reviewer -1. Our MS highlights the following new insights:

5. Rab22A has never been implicated in regulating the melanocyte pigmentation. This is the first study-reporting the Rab22A's role in melanogenesis.
6. Additionally, Rab22A showed an interaction with the well-known endosomal molecular complexes, BLOC-1 and BLOC-2, which regulate the pigmentation. This is the first study reporting that a GTPase regulates the function of BLOC-1 and BLOC-2.
7. Molecularly, Rab22A bridges BLOC-1 and KIF13A independently along with BLOC-2 to control the RE dynamics.
Additionally our study highlighted the following points:
8. Rab22A, Rab7A and Rab11A regulate the biogenesis of set of REs in which Rab22A is critically important for the dynamics of KIF13A or STX13-positive tubular REs.
9. Defects in RE dynamics observed in Rab22A-depletion phenocopied in BLOC-1 and BLOC-2-knockdown cells suggesting that BLOC-1/-2 functions downstream of Rab22A.

Editor P4: Moreover, both, referee 1 and 3 have a number of technical concerns

Response: We have rectified or answered all the technical concerns raised by the reviewers. Data related to their concerns included as Reviewer's figure (see at the end of this document).

Editor P5: and referee 1 indicated that the technical quality is low/unacceptable in the summary evaluation sheet returned with the report.

Response: We do not agree with the Reviewer-1. It seems that reviewer-1 is not aware of the complexity in working with self-assembled cytosolic complexes: BLOC-1, composed of 8 subunits and BLOC-2, encodes three subunits. Studying membrane association of these complexes biochemically is very difficult due to cytosolic nature of these subunits. Moreover, only 1 antibody each against the BLOC-1 and BLOC-2 are available with my colleagues in the field. Overexpression of any subunit of the complex may imbalance the complex architecture or its assembly on the membranes and thus, we used to detect the endogenous levels/interactions of these subunits. During the revision of this manuscript, we found two other BLOC-1 specific commercial antibodies (anti-dysbindin and anti-muted) and we have standardized the conditions and used along with anti-pallidin antibody. These two new antibodies supported the data generated from pallidin antibody. Thus, we request the Reviewer-1 to kindly look at our revised MS and our responses to judge the technical quality. If required, we will be happy to submit the raw data to the reviewers.

Editor P6: Given the nature of these concerns, the amount of work required to address them, the uncertain outcome of these experiments, and the fact that EMBO reports can only invite revision of papers that receive enthusiastic support from a majority of referees, I am sorry to say that we cannot offer to publish your manuscript.

Response: We have answered all the concerns raised by the reviewers. I request the editor to look at the revised manuscript and send it for external review.

Please see the responses to the reviewer's comments (point-by-point):

Referee #1:

In this manuscript, the authors focus on mechanisms of biogenesis of (tubular) recycling endosomes that both return internalized cargo to the plasma membrane and also, in certain cell types, direct cargo to lysosome-related organelles (such as melanosomes in melanocytes). The authors performed a limited shRNA screen for selected Rab proteins and calculated the number and length of tubular-shaped recycling endosomes as a measure of biogenesis. From this screen, the authors identified Rab22A as playing a significant role in recycling endosome biogenesis/morphology, and found that knock-down induced accumulation of cargo in endosomes or lysosomes. The authors provide evidence that Rab22A forms a complex with BLOC-1 and BLOC-2, as well as KIF13A, and that Rab22A may be required for the recruitment of the BLOC-complex proteins to the endosomal membranes.

Overall, this is an interesting manuscript that addresses important questions about the mechanism of recycling endosome biogenesis, and promotes a role for Rab22A in this process.

Response: We thank the reviewer to appreciate our work.

Reviewer-1 Q1: There are, however, serious concerns at both the conceptual and technical levels. First, based on the data shown and rationale given, it is unclear why the authors focused exclusively on Rab22 (in melanosomes).

Response: The rationale as follows:

3. We have chosen majority of the Rabs that are localized to the endosomal compartments in our RNAi screen excluding Rab7 (localizes to LEs) and Rab9 (LEs and lysosomes). Moreover, Rab7 (studies from others, Kawakami et al., 2008 and Hida et al., 2011) and Rab9 (our group, Mahanty et al., 2016) have been shown to regulate melanocyte cargo trafficking. But, none of the studies has shown the role of Rab22A in cargo transport to melanosomes or in pigmentation. During the revision, we have also evaluated the role of Rab7A, 8A, 10 and 21 in regulating RE biogenesis (see Reviewer's Fig.). Interestingly, we found Rab7A but not 8A, 10 or 21, regulates the KIF13A-positive tubules as similar to Rab11A and Rab22A. Further, we have extensively studied the role of each Rab on RE dynamics and maintenance (see new revised figures: Fig. 1, 4 and S1; reviewer's figure) and found that Rab22A plays a key role in RE biogenesis.
4. We have shown recently that BLOC-1, BLOC-2 and KIF13A independently regulate the transport of melanocytic cargo from REs to melanosome during their biogenesis (Setty et al., 2007, Dennis et al., 2015 and Delevoye et al., 2009). However, how these complexes are recruited to the membranes is unknown. We predicted that any of the endosomal Rabs possibly regulate the recruitment of these molecules and thus control RE dynamics. Small shRNA screen showed Rab7A, Rab11A and Rab22A potentially regulate the RE dynamics. Since, (1) Rab11A did not rescue the Rab22-dependent REs (Fig. S1A); (2) Rab7A did not localize to the Rab22A- or KIF13A-positive tubules (Fig. 1D and 1E); and (3) Rab7A and Rab11A did not show any biochemical interaction with Rab22A (Fig. 4E) or KIF13A (Fig. 4G), we predicted that Rab22A is the key to RE biogenesis. Consistently, new pull-down experiments clearly showed KIF13A interacts with Rab22A but not with Rab7/9/11 (Fig. 4G). Alternatively, Rab22A showed interaction with BLOC-1/-2 but not with Rab7/11 (Fig. 4E).

Reviewer-1 Q2: Second, there is a seeming disconnect between the stated goal of the manuscript and the actual description of the study. Initially the PI maintained that recycling endosome biogenesis in melanosomes was poorly understood, but the study addresses biogenesis in HeLa cells, where considerably more is known.

Response: We do not agree with the reviewer. Since, REs in any cell types follow the same mechanism. However, in melanocytes RE also carry melanocyte specific cargo along with non-melanocytic cargo. REs in melanocytes deliver the cargo toward melanosomes in addition to their cargo delivery to cell surface, similar to non-melanocytes. All our previous studies emphasized the importance of REs in melanosome biogenesis and pigmentation (Dennis et al., 2015, Jani et al., 2015 and Mahanty et al., 2016). Moreover, we can score the defects in RE function/ biogenesis easily in melanocytes than HeLa cells due to presence of melanin pigments. In this study we have shown the function of BLOC-1, BLOC-2 in HeLa cells, which was poorly appreciated previously. Thus, we do not have any disconnection between our goals. But we have used both HeLa and

melanocytes as a model to non-melanocytes and specialized cells respectively in this study. Our data suggest that the biogenesis mechanism of REs is similar in both the cell types.

Regarding the point on biogenesis of REs in HeLa cells: We do not agree with the reviewer. There are several types of tubular endosomes depending on the type of cargo or associated proteins localized to these tubules, discussed in the MS. However, their membrane origin can be variable. The REs described both in HeLa and melanocytes in this MS are derived from sorting endosomes (SEs), where BLOC-1, BLOC-2 and KIF13A known to function. Moreover, SE endosomes also generate another type of tubules dependent on Retromer (targets Golgi or cell surface) and their mechanism of biogenesis is very well known. Here we have described distinct recycling tubules that are originated from SEs but not from Golgi.

Reviewer-1 Q3: Indeed, a number of recent studies have addressed recycling endosome biogenesis and the various proteins and complexes involved, with emphasis on the tubular endosomes that resemble those observed by the authors for KIF13A, including studies by Takahashi et al., Wang et al., Bahl et al., Henmi et al., Xie et al., Compeer and Boes, Allison et al. Giridharan et al. and others.

Response: All recycling tubular structures morphologically looks similar to KIF13A/STX13-positive tubules. But these tubular structures are not the same. Since, most of the studies used internalized Tf as a marker to characterize these tubules. We predict that all these tubules can be labeled with internalized Tf due to the use of saturated levels of Tf in the experiments. Thus, tubules look very similar to KIF13A-positive tubules. Moreover, certain endogenous cargoes follow different types of tubules to reach their destination and this point was emphasized in the discussion of our MS.

Takahashi et al., - Study showed Rab11 localizes to REs and interacts with exocyst complex that regulates the exocytosis of REs toward plasma membrane. This study does not describe the biogenesis of REs or their origin or the mechanism.

Wang et al., - This study showed the role of Rab22 in NGF signaling and neurite outgrowth. Here Rab22A-Rabex5 interaction regulates the sorting of cargo rather than the biogenesis of RE. This study does not point out REs or their importance in the trafficking.

Bahl et al., - This study emphasize the EHD3 positive TRE (tubular recycling endosomes), possibly originated from Golgi and these are different than that of KIF13A-positive tubules. We have emphasized this point in the discussion of our manuscript. Our unpublished studies showed these tubular structures do not carry melanocytic cargo.

Henmi et al., We could not find the reference on PubMed.

Xie et al., - This study described the tubular recycling structures that contain cargo sorted either by CME or CIE pathway. Moreover, this paper emphasized the cargo sorting rather than the biogenesis of REs or the nature of REs.

Compeer and Boes, - These studies described the MICAL-L1-positive tubular structures similar to the TREs studied by Steve Caplan's group. We have described about these tubules in the discussion of our MS.

Allison et al. - This study describes that ESCRT-Spastin regulates the "fission" of REs rather than the biogenesis of REs.

Giridharan et al. - Several of his papers or Caplan's group showed the importance of adaptor EHD1 and their interaction with Rab8a and MICAL-L1 in endocytic recycling. All these molecules localized to the tubular structures and control either at endocytosis or tethering. We have discussed these points in the discussion. Moreover, our preliminary studies showed knockdown of Rab8A in HeLa cells had no effect on KIF13A-positive tubules (see Reviewer's Figure).

and others

NONE OF THESE STUDIES DESCRIBED THE MECHANISM OF BIOGENESIS OF REs or how they maintained the tubular characteristics. Our study detailed the role of BLOC-1/-2 in stabilizing the tubules with the help of Rab22A.

Reviewer-1 Q4: Even the role of Rab22A has been investigated in endosomes and recycling endosomes in non-melanocytes, undercutting, at least in part, the proposed rationale for this study.

Response: Reviewer is only looking at the known role of Rab22A rather than how this Rab controls the endosome maturation into REs. None of the studies so far reported how Rab22A regulates the cargo targeting from RE to cell surface without identifying the effector proteins. Reviewer was least considerate to know the mechanism (answered here) except the published literature on Rab22A.

Reviewer did not look at our new data that multisubunit complex BLOC-1 and BLOC-2 functions in

the Rab22A pathway to generate the REs. Thus, our MS did not compromise for any novelty in the work and the results are additive to strengthen the function of Rab22A.

Reviewer-1 Q5: In addition to these conceptual issues, as noted below, there are a variety of concerns regarding the data and its interpretation throughout the manuscript.

Response: We do not agree with the reviewer. If the reviewer has concern, we will be happy to provide the original data for his/her review (like Reviewer's figure). Our interpretations are very much suited to the results of the study. If the reviewer has specific point, we will be able to justify our interpretation.

Reviewer-1 Q6: It is difficult to accept the data presented in Figure 1. Rab5c is considered by the authors to cause cells to display significantly fewer tubules, with about 20% of the transcript remaining after shRNA transfection, whereas Rab5b supposedly does not display a significantly different number of tubules, and yet the expression of transcript after shRNA is at 85%. This makes no sense.

Response: We do not agree with reviewer that the entire data in Figure 1 has "no sense". Any RNAi screen, the knockdown efficiency of the genes varied depending on the potency of target sequences present in the siRNA/shRNA used. In our screen, except Rab5A and Rab5B, we have obtained more than 50% knockdown. As the reviewer might know that small ablation (either by over expression or knockdown) of Rab function can easily be scored to the phenotype. Thus, our phenotypes are much stronger as we showed in Figure 1 and we have quantified for the same. We have now repeated these experiments more carefully and obtained 57% knockdown with Rab5A pooled shRNA and 52% knockdown with Rab5B pooled shRNA, similar to other Rabs (Revised Fig. S1B). In these experiments, the localization and tubule length and number of KIF13A were not affected in Rab5A- or 5B-depleted cells compared to control cells [see Reviewer's figure for IFM data (5B not shown), similar to Fig. 1A]. Overall, our revised data clearly indicates that knockdown of either Rab5A or 5B had no effect on RE biogenesis. In contrast, Rab5C-knockdown moderately affected the tubule number and length, but not as significant as Rab7A (new molecule identified during revision) or 11A or 22A-depletion in the HeLa cells. Note, we observed enhanced cell death with stringent selection of Rab5A/5B/5C shRNAs (possibly increase the knockdown efficiency) and the cells are not in good/healthy condition to use it for any experiments.

Thus, we disagree with the reviewer that entire data has "no meaning". To our surprise, the other two reviewers have no concern on the screen. We request the reviewer to look at the preliminary screen done for other endosomal Rabs Rab8A, 10 and 21, showed no effect on KIF13A-positive tubules (Reviewer's figure).

Reviewer-1 Q7: In addition, KD efficacy should be measured at the protein level, and in any case, differences between Rab5a/b/c are miniscule and Rab5 protein levels should be addressed overall (same for the other Rabs).

Response: We agree with the reviewer that measuring protein levels in the knockdown cells provide better view in the case of isoform such as Rab5a/b/c. As reviewer pointed that there are no specific antibodies available to the Rab5 isoforms. Thus, we have shown the data in the form of transcript level, which is most acceptable form for the validation of RNAi screen. Moreover, the reviewer is making criticism more specific on Rab5 and missed the other important Rabs such as Rab11 and 22, which showed much more dramatic effect on RE dynamics. Additionally, we have obtained consistent results in Rab5 or its isoform knockdowns, which we repeated during the revision (Reviewer's figure, data not shown for Rab5B and 5C). If the reviewer feels that Rab5B and 5C are not important for the study, we will be happy to remove it from the MS.

Reviewer-1 Q8: How does Rab9 fit into the pathway? The authors previously implicate Rab9 in this pathway, but fail to discuss its role and why it differs from Rab22A.

Response: We have not considered Rab9 (reduction in the number of tubules is not significant) initially, even though we have shown its role in cargo transport to melanosomes. To remind the reviewer that we have implicated Rab9 during the fusion of REs with melanosome rather than RE biogenesis. Reviewer missed this point from our previous studies. Moreover, Rab9 in non-melanocytes majorly functions between Golgi-lysosome pathway.

To answer the reviewer's point, we have performed several experiments to determine the role of Rab9A in RE biogenesis: (1) cohort of Rab9A was localized to the STX13-positive endosomes (Fig.

S1D), but not to the KIF13A-positive tubules (Fig. S1E); (2) overexpression of Rab9A had no effect on KIF13A-tubular structures (Fig. S1E) and they are similar to empty vector transfected cells (Reviewer's figure); (3) GFP-Rab9A localized as punctate structures and did not show any colocalization with Rab22A-tubules by live cell imaging (Fig. S1F) and (4) biochemically Rab9 showed no interaction with KIF13A (Fig. 4G). These results clearly demonstrated that Rab9A is not associated with Rab22A-BLOC-1-BLOC-2-KIF13A pathway. Consistently, Rab9A knockdown has no effect on RE tubule length but moderately affected the tubule number in HeLa cells. This data indicates that Rab9A is functionally different from Rab22A in regulating RE dynamics. These points were included in the revised manuscript.

Reviewer-1 Q9: Is the pathway controlled dually by Rab22 and Rab9?

Response: We do not agree with the reviewer's hypothesis. Both have different functions. Our previous studies (Mahanty et al., 2016) showed Rab9 functions during the fusion of REs with melanosomes with the help of Rab38/32/VARP, which regulates the STX13 and VAMP7 fusion localized on respective membranes. Whereas, Rab22A (in the current study) localizes to REs and SEs but not to the melanosomes. Our data suggests that RE biogenesis from SEs possibly regulated by Rab22 and Rab11 rather than Rab9A (also see the point: Reviewer-1 Q8).

Reviewer-1 Q10: In Fig. 1D, the authors claim that the Rab22A Q64L mutant displays significantly enhanced localization to KIF13A recycling tubules, yet the numbers show a 0.48 as opposed to 0.42 Pearson's Coefficient. By what basis is this "significant?"

Response: We agree with the reviewer's concern on Pearson's coefficient values. We would like to point to the reviewer that 'r' values are best for endocytic vesicles rather than tubular structures. To evaluate the role of Rab22A constitutive mutant in enhancing the KIF13A membrane localization, we have quantified the Corrected Total Cell Fluorescence (CTCF) in both Rab22A^{WT} and Rab22A^{Q64L} conditions and then plotted (see new Figure S1G). Interestingly, we observed slightly enhanced fluorescence (1.13 folds) in cells expressing Rab22A^{Q64L} (1.56×10^8 A.U.) compared to Rab22A^{WT} (1.38×10^8 A.U.) expressing cells. Likewise, we observed modestly increased fluorescence (1.57 folds) of KIF13A in Rab22A^{Q64L} (1.10×10^8 A.U.) compared to Rab22A^{WT} (0.70×10^8 A.U.) expressing cells (Fig. S1G). In line, the increased CTCF values for Rab22A^{Q64L} and KIF13A also correlate to the moderately increased Pearson's coefficient value (r) (Fig. 1F) between these molecules compared to Rab22A^{WT} and KIF13A. Thus, overexpression of Rab22A^{Q64L} slightly enhances localization of KIF13A compared to Rab22A^{WT}. Consistently, knockdown of Rab22A showed moderately reduced localization of KIF13A to the membranes (Fig. 2A for IFM and Fig. S2B for biochemical analysis). Overall, this data suggests that overexpression of Rab22A modestly increased the membrane localization of KIF13A. We have included this data in the revised manuscript.

Reviewer-1 Q11: How do the authors know that overexpressing the Rab22A mutant doesn't alter KIF13A localization?

Response: We thank the reviewer for nice question. We did not observe any major alteration (i.e., destabilization) in KIF13A membrane localization (based on IFM studies) both in Rab22A^{WT} and Q64L mutant overexpressing cells (Fig. 1F) compared to empty vector expressing cells (see Reviewer's figure). In contrast, Rab22A^{Q64L} mutant moderately enhances the KIF13A membrane localization compared to Rab22A^{WT} (based on CTCF analysis shown in Fig. S1G). This is the probable reason for increased number of tubules (but not the tubule length, see Fig. 1H) upon overexpression of Rab22A^{Q64L} mutant compared to Rab22A^{WT} (Fig. 1G) in HeLa cells. Consistently, overexpression of Rab22A^{S19N} (Fig. 1F) or the knockdown of Rab22A (Fig. 2A) in HeLa showed slightly reduced KIF13A membrane association (only restricted to SEs). These results are consistent with a weak reduction in membrane association of endogenous KIF13A (Fig. S2B). Thus, KIF13A localization is mildly altered i.e., enhanced with the overexpression of Rab22A (WT or Q64L).

Reviewer-1 Q12: I have concerns about the authors' use of the term RE dynamics when they are looking at fixed cells.

Response: For reviewer's note, we have used maximum intensity projection of the Z-stack to quantify the dynamics rather than single frames of cells. Moreover, we have observed similar kinetics in live imaging (data not shown). If needed, we will be happy to provide the live images of the same.

Reviewer-1 Q13: Figure 4A and B is not compelling. The 8% decrease in membrane association by pallidin upon Rab22A-depletion is (Fig. 4B) not convincing, and therefore, the conclusions about membrane association in Fig. 4A are similarly problematic.

Response: We completely agree with the reviewer and thanks for his/her comment. Reviewer should also note that BLOC-1 contains 8 subunits, however, knockdown of one subunit destabilizes the entire complex (Setty et al., 2007, Dennis et al., 2015). Thus, we have used Muted subunit of BLOC-1 complex for the knockdown in HeLa cells to represent the BLOC-1-deficient cells. Similarly, melan-mu melanocytes derived from Muted mice were used. However, there are no very good antibodies against the BLOC-1 subunits. Between the available 2 antibodies [anti-Muted (available with Prof. Mickey Marks, CHOP-UPENN, Philadelphia) and anti-Pallidin (Prof. Esteban Dell'Angelica, UCLA, USA)] only anti-pallidin gives specific band compared to others. Thus, we have used anti-pallidin antibody for most of the experiments previously. During revision, we extensively searched for antibodies against different subunits of BLOC-1. Luckily, we found two new antibodies: anti-dysbindin (from Abcam, ab118795) and anti-Muted (from Sigma-Aldrich, SAB1303447) and evaluated them using respective knockdown cell lysates (data not shown) and standardized the conditions. These antibodies work at very low dilution with minimal non-specificity.

Using these antibodies, we have repeated the subcellular fractionation (new Fig. 4A), membrane-cytosol association studies (new Fig. 4B) and pull-down experiments (new Fig. 4D) and revised the entire Figure 4. In these experiments, we probed for at least two different subunits (Pallidin, Dysbindin or Muted) of BLOC-1 (previously we probed only one subunit, Pallidin) to study the membrane association of BLOC-1 complex. The new subcellular fractionation experiment in Fig. 4A showed all three subunits were critically reduced their levels and membrane association upon Rab22A-knockdown in HeLa cells. Consistently, the new membrane-cytosol association experiment in Fig. 4B showed 28% reduction in membrane association of Pallidin upon depletion of Rab22A. As expected, the membrane association of dysbindin followed the pallidin subunit (Fig. 4B), suggesting that Rab22A regulates the membrane association of BLOC-1. Additionally, Rab22A showed interaction with BLOC-1 (pallidin and muted) subunits, shown in revised Fig. 4D. Thus, the data in Fig.4A is inline with Fig.4B. Moreover, we have performed additional experiments to identify the mechanism between Rab22A, BLOC-1, BLOC-2 and KIF13A using pull-down experiments (new Fig. 4G). In these experiments, KIF13A showed interaction with Rab22A and BLOC-2, but not with BLOC-1, suggesting that Rab22A bridges the BLOC-1 and KIF13A independently. Additionally, BLOC-2 interacts with both Rab22A and KIF13A. Based on this data, we have revised the model and present in Fig. 4H. Overall, the new results of revised Fig. 4A and 4B are consistent with each other.

Note that these experiments are tedious and very difficult in nature. Hope, the revised experiments and additional pull-down experiment convince the reviewer to appreciate our effort in identifying the mechanism.

Reviewer-1 Q14: Fig. 4D is not compelling. The smeared band for Muted is not convincing.

Response: We completely agree with reviewer. Thank you for pointing this important point. We have carefully analyzed the original data (enclosed in the Reviewer's figure) and found our mistake that we have highlight the wrong sized band (non-specific) for Muted. Since, the expected size for Muted is around 25 kDa. We found strong high intense band in the His-Rab22A pull-down lane. Due to saturation, we also observed a very low intense band in the control lane. However, we have repeated this experiment and added the new data in Fig. 4D.

Reviewer-1 Q15: How do the authors define "strongly" when they note that Rab22A strongly binds to the BLOC-1 complex?

Response: The prediction is based on Fig. 4D and 4E results, where the interaction between Rab22A and Pallidin/Muted showed stronger and followed Rab nucleotide dependency.

MINOR:

Reviewer-1 Q16: Why do the control shRNA HeLa look so different in 1A and 2A for KIF13A?

Response: The cell showed in previous Fig. 2A is a dividing cell and looks slightly different than Figure 1A. We have replaced this figure with new one.

Reviewer-1 Q17: Can endogenous KIF13 be studied to give more uniform results?

Response: Antibody does not work for immunostaining other than the blotting (works at very low dilution). Moreover, YFP-KIF13A does not express in melanocytes (due to low transfection efficiency). Thus, we have expressed this motor in HeLa cells wherever it is required.

Reviewer-1 Q18: This reviewer doesn't see any tubules in the sh control, yet the average number of tubules per cell in the graph is listed at 25.

Response: We do not agree with the reviewer on this point. We have observed the tubules and emphasized as inset in sh control panel (Fig. 1A). We request the reviewer to see the similar YFP-KIF13A tubules in Fig. 1D, Fig. 2A, Fig. 2E, Fig. S1A, Fig. S1E, Fig. S2A and reviewer's figure. We have used the algorithm (see materials and methods) to quantify the tubules using Image J. We request the reviewer to look at the arrowheads indicated in the listed above figures.

Reviewer-1 Q19: The manuscript needs to be better edited for clarity, and has numerous sentences that are written awkwardly and are difficult to follow. One example of many: "Moreover, the pigment content of the Rab22A-depleted melanocytes was almost as equal as in..."

Response: We have corrected and edited the text at respective places.

Referee #2:

The manuscript by Shakya et al entitled "Rab22A brings BLOC-1 and BLOC-2 to promote the biogenesis of recycling endosomes" presents evidence that Rab22A, BLOC-1 and BLOC-2 are important for the formation of tubular recycling endosomes that are used for cargo recycling and delivery of cargo to melanosomes. They begin by looking for the Rab that is important for the generation of tubular recycling endosomes marked by overexpressed KIF13A. Surprisingly a number of endosomal Rabs led to a noticeable reduction of KIF13A tubules but they focused on Rab22A since it had the strongest phenotype. They show that loss of BLOC-1 or BLOC-2 also leads to loss of tubular endosomes and then show that Rab22A binds to BLOC-1, BLOC-2 and KIF13A and that Rab22A may be responsible for the recruitment of the BLOCs to the membrane. Loss of Rab22A has a pronounced phenotype on cargo transport to melanosomes, resulting in reduced pigmentation. This is an interesting finding that places Rab22A at a junction influencing endosomal membrane recycling and also traffic to the melanosome.

We thank the reviewer for the summary of our work and his/her appreciation.

There are only a few issues that should be addressed (that shouldn't involve additional experiments).

Reviewer-2 Q1: It is mentioned in the text, p. 3 that overexpression of Rab11A did not rescue the loss of KIF13A tubules observed in Rab22A-depleted cells. Since Rab11-depleted cells also leads to loss of tubules, did Rab22A overexpression rescue this phenotype?

Response: We thank the reviewer for nice comment. We have performed this experiment (See Reviewer's figure) and found that Rab22A partly rescued the YFP-KIF13A tubules in Rab11A-knockdown cells. However, the tubule length in Rab11A-depleted cells appeared to be shorter compared to control cells. Interestingly, membrane association of YFP-KIF13A in Rab22A expressing Rab11A sh cells is almost equivalent that of control cells and increased compared to the Rab11A-alone (Fig. 1A). Since, Rab11A has a well-known function at recycling endosomes; and thus we did not focus on this Rab during our initial experimentation.

Reviewer-2 Q2: Also, how do the authors interpret the overall loss of KIF13A tubules in many of the Rab depletion experiments?

Response: We have observed reduced tubule number in Rab7A- (new Rab, identified during revision), Rab11A- and Rab22A-knockdown cells. However, we have previously shown that Rab9A is involved in targeting REs to melanosomes (Mahanty et al., 2016); and thus we have used this Rab for further analysis. To understand the role of different Rabs on YFP-KIF13A-positive tubule biogenesis, we have colocalized these Rabs with KIF13A (see new Fig. 1D). Extensive IFM analysis showed Rab11A and Rab22A were localized majorly to KIF13A-positive tubules compared to Rab7A and Rab9A (Fig. 1D and S1E). Live cell-imaging analyses further showed that Rab22A form longer tubular structures that are partly positive for Rab11-punctate structures (Fig. 1E). Interestingly, Rab7A colocalized with Rab22A punctate structures but not tubular endosomal

structures, where as Rab9A does not show any colocalization with Rab22A-tubular structures (Fig. 1E and S1F). Thus, Rab22A is a potential candidate for regulating KIF13A-positive tubules. However, the loss of KIF13A-tubules in Rab7A- and Rab11A-knockdown cells might be due to the altered endosomal structures/organization.

Reviewer-2 Q3: In all cases the KIF13A appears to be trapped out in the periphery.

Response: These peripheral vesicles or clusters of sorting endosomes that are positive for internalized Tf, AP-1 (Fig. 2A), Rab11A (Fig. S1A) and STX13 (not shown).

Reviewer-2 Q4: The quantification of the western blots in Fig. 3C and F might be worth a second look. There is little change reported for HPS6 in C but it looks by eye to be a bigger change; the large change reported for Rab5 expression in F seems to be minor by eye.

Response: We thank the reviewer for noticing such differences. We re-quantified the data with their respective tubulin levels (now included in the figure) and modified changes (values) in the figures (Fig. 3C and 3F). Now, we observed HPS6 levels were not altered upon Rab22A-knockdown in melanocytes, whereas Rab5 level were modestly altered. We agree with the reviewer that the appeared bands look different, but the quantification with their respective internal controls showed the difference, which we have highlighted in the text.

Reviewer-2 Q5: Weigert et al (2004 - ref #20) was the first to report that loss of Rab22A led to loss of recycling tubules in HeLa cells, suggesting that Rab22A promoted tubule formation from SEs. This study should be cited or acknowledged in Results/Discussion.

Response: We agree with the reviewer and included in the text at respective places.

Referee #3:

The study by Shakya and colleagues details experiments to investigate the process of formation of recycling endosomes (REs) and reports a role for Rab22a acting, at least in part, through the BLOC-1 and BLOC-2 complexes. REs are important elements of the endocytic pathway being involved in recycling proteins to the cell surface. They also contribute to trafficking of proteins to lysosome-related organelles (LROs). There is a lot known about much of the machinery involved in endosomal protein sorting although the formation of REs remains somewhat mysterious and there is certainly a case for studying how REs are formed.

The data presented in the manuscript by Shakya and colleagues does indicate that Rab22a has an important role to play in RE formation/dynamics and this represents an advance in the understanding of REs formation. I do think however that the manuscript needs substantial revision before it is suitable for publication in EMBO Reports.

We thank the reviewer for the summary. We have extensively revised the manuscript with additional experiments.

Major issues:

Reviewer-3 Q1: The manuscript mentions at the start that REs are functionally connected to the TGN but afterwards the role of the TGN (and associated machinery) appears to be completely overlooked. AP1 is important in RE formation/function and may do so by interacting with Rab4 and Rab5 regulating proteins - could Rab22a be regulated through the same proteins that associate with AP1?

Response: AP-1 is a known adaptor for cargo sorting both at the TGN and endosomes and it does through Rab4 and Rab5. Our revised results showed AP-1 did not show any interaction with Rab22A biochemically (Fig. 4D and 4E), even though AP-1 (1) localizes to the Rab22A-positive membrane fractions by subcellular fractionation (Fig. 1I) and (2) localizes to SEs (positive for YFP-KIF13A) in the Rab22A-knockdown cells. Interestingly, AP-1 showed a strong interaction with tail domain of KIF13A (Fig. 4G), suggesting that AP-1 functions in the pathway, possibly upstream of Rab22A. These results were consistent with the previous study carried out by the co-author, suggesting that AP1-KIF13A is required for the RE generation (Delevoeye et al., 2009). Thus, our study supports that Rab22A extends the vesicle buds into tubular structures but may not act during the sorting of cargo. We have discussed about different tubular structures including the ones derived from Golgi in discussion of the MS.

Reviewer-3 Q2: There does appear to be some changes in AP1 localization after loss of Rab22a expression but the contribution of AP1 to RE formation does not really get much coverage in the manuscript.

Response: We predict and our data suggest that AP-1 functions upstream in the Rab22A pathway during cargo sorting, but may not regulate the RE formation. Co-author of this MS already showed AP-1-KIF13A regulates the RE formation (Delevoeye et al., 2009). Studies here are more focused on the down stream of AP-1, where AP-1 and KIF13A already localized to the membranes on which Rab22A-BLOC-1-BLOC-2 promote the extension of the tubules. We have indicated this point in the discussion and due to limitation in word count (short article), we have not elaborated the AP-1 function in RE biogenesis.

Reviewer-3 Q3: In the introduction the authors list some of the endosomal protein sorting machinery such as retromer and the WASH complex and make distinctions between different sets of proteins. There is however ample evidence that the sets of machinery can interact indirectly or regulate each other. For example, it has been reported that the WASH complex not only associates with retromer but also with the BLOC1 complex.

Response: We agree with the reviewer. Co-author of this MS already showed BLOC-1- WASH complex is possibly required for the tubule stabilization or fission of the RE (Delevoeye et al., 2016). We predict that these molecules function down stream of Rab22A-BLOC1-BLOC-2-KIF13A complex.

Reviewer-3 Q4: Another retromer accessory protein, VARP has a role in trafficking to LROs and retromer proteins have been shown to associate with proteins involved in RE tubule formation/maintenance such as EHD1. I think some changes to the introduction would be worthwhile so that the impression that is not made that the various endosomal protein sorting machineries operate in total isolation.

Response: We agree with the reviewer. More details of different types of tubular structures and their machinery can be included in the text. Due to word limit (short article), we were much focused in explaining the different components of our complex Rab22A-BLOC-1-BLOC-2-KIF13A. In this revision, we have included different types of tubular structures and their regulated machineries in the discussion.

Reviewer-3 Q5: In Figure 1, the authors describe their tests for Rab involvement in regulating RE tubule number/length as a "screen". I think it is an overstatement to call it a screen when only 8 different rab proteins were tested.

Response: We have changed the sentence to “selected Rab shRNA screen”.

Reviewer-3 Q6: What was the rationale in choosing those eight proteins?

Response: We have chosen the Rabs that localize to the EE, SE and RE. We have avoided late/lysosomal or melanosome localized Rabs.

Reviewer-3 Q7: Why wasn't Rab7a, Rab8, Rab6 and Rab10, Rab21 included in the "screen?"

Response: We thank the reviewer for his /her suggestion. We have carried out preliminary screening of YFP-KIF13A-tubule distribution (membrane bound or cytosolic) in these Rabs knockdown cells (see Reviewer's figure). Here, we have knockdown the cells with their respective pooled shRNAs and selected them for stringent Puromycin selection (shRNA plasmid selection), transfected with YFP-KIF13A and visually counted the cells for KIF13A tubule distribution. As a control, Rab5A was used and observed only 69% of cells showed KIF13A-tubules (rest showed cytosolic KIF13A) compared to control cells (92% of cells showed tubules). Interestingly, Rab7A-knockdown displayed only 31% of cells having tubules, whereas Rab8A, 10 and 21-depletion showed no-change in tubule distribution (similar to control). We have chosen Rab7 for further studies and included the data in the main figure 1. Moreover, we have extended our analysis how Rab7, Rab9, Rab11 and Rab22A differ in regulating the RE dynamics (see Fig. 1 and Fig. S1).

Reviewer-3 Q8: In the tubule counting/measuring experiments, how were the cells selected for imaging? Was the imaging done blind to avoid potential bias?

Response: We appreciate the reviewer point. We have taken all the cells expressing YFP-KIF13A. However, we have quantified the tubule length/number in around 15 cells/condition (except Rab14 sh) and indicated in the graph. Thus, we have followed completely blind study during the quantification.

Reviewer-3 Q9: In figure 1G, it is hard to conclude that the proteins being investigated cofractionate when so few fractions are analyzed and no marker proteins for the TGN or sorting endosomes or lysosomes are shown on the blot. Do the fractions correspond to the steps for the 'gradient' used?

Response: We have included the other markers as suggested in the new Fig. 1I. The fractions correspond to the sucrose gradients as indicated.

Reviewer-3 Q10: In figure 2D, the quantification of the blot looks rather odd. For example, the STX13 signal in lane 3 (BLOC1sh) is apparently twice as much as lane 1 (control) but looks very similar even when you factor in the tubulin loading control. I suspect that the rather variable grey background in the different blots may have influenced the quantification.

Response: We completely agree with the reviewer. We have repeated the experiment, quantified the blots and included in the figure.

Reviewer-3 Q11: In the blot shown in figure 1, STX13 is a doublet but only a single band in Figure 2 - why so?

Response: We thank the reviewer for this point. We have used anti-STX13 from Andrew Peden's lab in Figure 1, where the antibody highlights two bands. In contrast, Abcam antibody detects single band, used in the previous Figure 2. For consistency, we have repeated the experiments with anti-STX13 antibody from Peden's lab (gives doublet) and included the data in the revised figures.

Reviewer-3 Q12: In figure 4D or E (or both) I would like to see a coomassie stained gel of the pull-down so that it can be judged how clean/dirty the pull-downs are.

Response: We have included the full-length coomassie stained gels of Fig. 4D and Fig. 4E for the reviewer in reviewer's figure.

Figure for reviewers:

Reviewer-1, Q11: Effect of Rab22A on KIF13A membrane localization. Cells were co-transfected with YFP-KIF13A and empty vector (pCDNA3.1) or Rab22A, methanol fixed and then imaged.

Reviewer-1, Q14: Original gel of His-Rab22A pull-down using HeLa cells. We have highlighted the non-specific band as Muted in our previous Fig. 4D. This is for reviewer's information.

Reviewer-2, Q1: Rab22 overexpression partially compensate the loss of YFP-KIF13A-tubules in Rab11A-knockdown cells. Indicated cells were transfected, fixed and then imaged.

Reviewer-3, Q7: Role of Rab7A, Rab8A, Rab10 and Rab21 in regulating KIF13-tubule distribution. Indicated cells were transfected with YFP-KIF13A, methanol fixed and then imaged randomly. Cells were counted for KIF13A membrane or cytosolic distribution and listed. Rab5A used as control, similar to Fig. 1A.

Reviewer-3, Q12: Full length coomassie gels of His-Rab22A pull-downs using HeLa and melanocytes lysates respectively. This is for reviewer's information.

Thank you for the submission of your revised manuscript. I have taken over its handling as my colleague Martina is currently out of the office. We have now received the enclosed reports from referees 2 and 3 plus comments from an advisor who assessed how well referee 1's concerns were addressed. I am happy to tell you that the referees support the publication of your study now.

However, from the advisor's comments it is clear that the manuscript needs some extensive re-writing to increase clarity. The manuscript will be published as a short scientific report, given the 4 main figures, however, the text should be shortened to below 30.000 characters (including spaces and figure legends but excluding materials and methods and references) or at least as much as possible. Re-writing the text to increase clarity and at the same time shorten the text will be the main task.

Apart from this there are several things we need from the editorial side.

REFEREE REPORTS

Referee #2:

The study provides evidence that Rab22A together with BLOC-1/BLOC-2 and Kif13A are involved in endocytic recycling tubule formation and transport to the melanosome.

Referee #3:

The revised manuscript has addressed the concerns I expressed in my previous review. Although this view may not be in line with the other reviewers, I believe that the manuscript is now appropriate for publication in EMBO Reports.

Advisor's comments:

Although the reviewer's concerns are reasonable, the authors do not always provide adequate answers in the rebuttal letter. As a consequence, the logic is often convoluted and not easy to follow. For example reviewer one asks to clarify how Rab9 fits into the pathway (Q8). Authors reply that Rab9 in melanocytes is involved in fusion of endosomes (REs) with melanosomes and not in RE biogenesis. And they add that in other cells (non melanocytes) Rab9 functions in Golgi-endosome transport. However, in the paper, they use HeLa cells! The issue raised by the reviewer is legitimate and the answer is inappropriate. However, this is not a main concern.

In the end, I believe that the authors have addressed the main concerns of the first reviewer - but I have to admit that I am not fully convinced by some of the Figs. Also the paper itself is not well written, or awkwardly written (reviewer one Q19). The authors should have it read/corrected by colleagues.

In fact, I realise that my problem comes primarily from the text - and I guess that this was the main problem of reviewer one. Indeed, much of his/her comments raise issues that need to be clarified in the text (e.g. Rab proteins). This is also the way I feel, and I think that the data are basically fine.

Point-by-point responses to your suggestions/comments:

Point-1: Dear Prof. Gangi Setty,

Thank you for the submission of your revised manuscript. I have taken over its handling as my colleague Martina is currently out of the office. We have now received the enclosed reports from referees 2 and 3 plus comments from an advisor who assessed how well referee

1's concerns were addressed. I am happy to tell you that the referees support the publication of your study now.

Response: Many thanks for handing our manuscript during revision. We "THANK" you and Martina for your support throughout the review process.

Point-2: However, from the advisor's comments it is clear that the manuscript needs some extensive re-writing to increase clarity.

Response: We have revised our text to increase the clarity

Point-3: The manuscript will be published as a short scientific report,

Thanks so much for considering as Scientific Report.

given the 4 main figures,

We have divided the Fig 1 into new Fig 1 and Fig 2. Now, the revised MS contains five main figures and four Appendix figures.

however, the text should be shortened to below 30.000 characters (including spaces and figure legends but excluding materials and methods and references) or at least as much as possible. Re-writing the text to increase clarity and at the same time shorten the text will be the main task.

We have reduced the MS text size to the maximum extent. However, the current text is slightly higher than 30.000 characters. I kindly request you to allow us to publish with the current text length.

Point-by-point responses to the referee comments:

Referee #2:

The study provides evidence that Rab22A together with BLOC-1/BLOC-2 and Kif13A are involved in endocytic recycling tubule formation and transport to the melanosome.

We thank the Reviewer-2 for his/her suggestions and kind support throughout the manuscript revision.

Referee #3:

The revised manuscript has addressed the concerns I expressed in my previous review. Although this view may not be in line with the other reviewers, I believe that the manuscript is now appropriate for publication in EMBO Reports.

We thank the Reviewer-3 for his/her critical comments/suggestions and kind support throughout the manuscript revision.

Advisor's comments:

Although the reviewer's concerns are reasonable, the authors do not always provide adequate answers in the rebuttal letter. As a consequence, the logic is often convoluted and not easy to follow. For example reviewer one asks to clarify how Rab9 fits into the pathway (Q8).

Authors reply that Rab9 in melanocytes is involved in fusion of endosomes (REs) with melanosomes and not in RE biogenesis. And they add that in other cells (non melanocytes) Rab9 functions in Golgi-endosome transport. However, in the paper, they use HeLa cells! The issue raised by the reviewer is legitimate and the answer is inappropriate. However, this is not a main concern.

In the end, I believe that the authors have addressed the main concerns of the first reviewer - but I have to admit that I am not fully convinced by some of the Figs. Also the paper itself is not well written, or awkwardly written (reviewer one Q19). The authors should have it read/corrected by colleagues.

In fact, I realise that my problem comes primarily from the text - and I guess that this was the main problem of reviewer one. Indeed, much of his/her comments raise issues that need

to be clarified in the text (e.g. Rab proteins). This is also the way I feel, and I think that the data are basically fine.

We thank the Advisor for his/her critical suggestions, which helped us in rewriting the part of the text. We immensely thank him/her for supporting our MS at critical time of manuscript review.

3rd Editorial Decision

15 August 2018

Thank you for your patience while we have gone through your revised manuscript and also for incorporating all requested changes. I am therefore writing with an 'accept in principle' decision, which means that I will be happy to accept your manuscript for publication once a few minor issues/corrections have been addressed, as follows.

- My colleague Esther Schnapp asked you to shorten and rewrite the manuscript not only to get the character count closer to that of Scientific Reports (25,000 plus/minus 2000 characters) but also to make the text more concise and clearer. I notice that you shortened the manuscript a little bit but that it overall remained rather unchanged. It is not necessary to cut it down to 27,000 characters, if it is close to 30,000/31,000 (introduction, results and discussion) this will be sufficient but it would be good to increase conciseness and clarity as also outlined by our advisor. I have taken the liberty to rewrite the first two paragraphs a little bit (please see attached document).

Please note that it is no problem to include five figures as EV figures. The EV figure legends are not included in the word count. Therefore, please feel free to promote five figures of your choice to the Expanded View content.

- I noticed that the experiment shown in Fig. S1H was only done once (if I understand correctly). I therefore kindly ask you to remove the statistical analysis in this panel.

- In Figure 4D you show EM pictures. The upper panel shows an overview and the lower panel "insets". Are these truly insets derived from the overview images or higher resolution images derived from another set of EM scans? if the first option is true, please indicate where the insets come from, if the latter option is true, it might make sense to call these "higher magnification images" or alike.

If all remaining corrections have been attended to, you will then receive an official decision letter from the journal accepting your manuscript for publication in the next available issue of EMBO reports. This letter will also include details of the further steps you need to take for the prompt inclusion of your manuscript in our next available issue.

3rd Revision - authors' response

30 August 2018

Point-by-point responses to your suggestions/comments:

Point-1: Dear Subba,

Thank you for your patience while we have gone through your revised manuscript and also for incorporating all requested changes. I am therefore writing with an 'accept in principle' decision, which means that I will be happy to accept your manuscript for publication once a few minor issues/corrections have been addressed, as follows.

Response: Many thanks for your wonderful decision. We have addressed all the minor issues/corrections listed below and included all changes in the revised text.

Point-2: My colleague Esther Schnapp asked you to shorten and rewrite the manuscript not only to get the character count closer to that of Scientific Reports (25,000 plus/minus 2000 characters) but also to make the text more concise and clearer. I notice that you shortened the manuscript a little bit but that it overall remained rather unchanged. It is not necessary to cut it down to 27,000 characters, if it is close to 30,000/31,000 (introduction, results and discussion) this will be sufficient but it would be good to increase conciseness and clarity as also outlined by our advisor.

Response: Thank you and Dr. Esther for the suggestion on rewriting the text to increase the clarity. I have rewritten all most all parts of the results and discussion section and few parts of the introduction. Now, I am happy to say that the current text is much more

understandable than the previous version. Please look into the text and let me know, if I need to make any further changes. The current text length contain lesser than 25241 characters (introduction, results and discussion). Hope, it is acceptable to you.

Point-3: I have taken the liberty to rewrite the first two paragraphs a little bit (please see attached document).

Response: Many thanks for edits. It is a great help and support and I followed your writing pattern through out the text.

Point-4: Please note that it is no problem to include five figures as EV figures. The EV figure legends are not included in the word count. Therefore, please feel free to promote five figures of your choice to the Expanded View content.

Response: Thanks so much for allowing us to modify the Appendix figures into EV figures. Now, we have included them as EV figures.

Point-5: - I noticed that the experiment shown in Fig. S1H was only done once (if I understand correctly). I therefore kindly ask you to remove the statistical analysis in this panel.

Response: Thanks for your suggestion. You are absolutely correct that the data values are from two independent experiments (not three). Thus we removed the statistics from the Fig EV1H (old Fig S1H).

Point-6: - In Figure 4D you show EM pictures. The upper panel shows an overview and the lower panel "insets". Are these truly insets derived from the overview images or higher resolution images derived from another set of EM scans? if the first option is true, please indicate where the insets com from, if the latter option is true, it might make sense to call these "higher magnification images" ore alike.

Response: Many thanks for your nice comment. We have indicated (as black boxes) the source of magnified inset on larger image. Few of the insets are from different EM scans of representative condition. These points are included in the legend of Figure 4 of revised text.

Point-7: Once you have made these minor revisions, please use the following link to submit your corrected manuscript:

<https://embor.msubmit.net/cgi-bin/main.plex?el=A11j1BWH7A2CcPu5J1A9ftdzaJivxtn2Z0BuaMih5oKXgY>

Response: Thanks so much for your suggestions/comments.

Point-8: If all remaining corrections have been attended to, you will then receive an official decision letter from the journal accepting your manuscript for publication in the next available issue of EMBO reports. This letter will also include details of the further steps you need to take for the prompt inclusion of your manuscript in our next available issue.

Thank you for your contribution to EMBO reports. Kind regards,
Martina

Response: We have included all the suggestions/corrections suggested by you in the revised manuscript. Hope, we will hear a positive final decision of acceptance from you soon. Once again thank you very much for all your "support" through out the revision time of our manuscript.

Thank you for your mail. I am now writing again with an 'accept in principle' decision, which means that I will be happy to accept your manuscript for publication once a few minor issues/corrections have been addressed, as follows.

- You noticed a labeling mistake in some figures (KIF13A-YFP versus YFP-KIF13A). Please

update the respective figures and upload new versions.

- Please also upload the revised manuscript with an extended discussion on the interaction data (Rab11, KIF13A).

4th Revision - authors' response

1 October 2018

The authors performed all minor editorial changes.

Corresponding Author Name: Subba Rao Gangi Setty

Manuscript Number: EMBO-2018-45918V5